# OmniVinci: Enhancing Architecture and Data for Omni-Modal Understanding LLM

Hanrong Ye[†*]    Chao-Han Huck Yang[*]    Arushi Goel[*]    Wei Huang[*]    Ligeng Zhu[*]
Yuanhang Su[*]    Sean Lin[*]    An-Chieh Cheng[*]    Zhen Wan[*]    Jinchuan Tian[*]    Yuming Lou[*]
Dong Yang[*]    Zhijian Liu    Yukang Chen    Ambrish Dantrey    Ehsan Jahangiri    Sreyan Ghosh
Daguang Xu    Ehsan Hosseini-Asl    Danial Mohseni Taheri    Vidya Murali    Sifei Liu    Yao Lu
Oluwatobi Olabiyi    Yu-Chiang Frank Wang    Rafael Valle    Bryan Catanzaro    Andrew Tao
Song Han    Jan Kautz    Hongxu Yin[§†*]    Pavlo Molchanov[§]

## NVIDIA

[*]Core Contribution    [†]Corresponding Authors    [§]Equal Advisory

**Code    Model    Webpage**

## Abstract

Advancing machine intelligence requires developing the ability to perceive across multiple modalities, much as humans sense the world. We introduce OmniVinci, an initiative to build a strong, open-source, omni-modal LLM. We carefully study the design choices across model architecture and data curation. For model architecture, we present three key innovations: (i) OmniAlignNet for strengthening alignment between vision and audio embeddings in a shared omni-modal latent space; (ii) Temporal Embedding Grouping for capturing relative temporal alignment between vision and audio signals; and (iii) Constrained Rotary Time Embedding for encoding absolute temporal information in omni-modal embeddings. We introduce a curation and synthesis pipeline that generates 24M single-modal and omni-modal conversations. We find that modalities reinforce one another in both perception and reasoning. Our model, OmniVinci, improves over Qwen2.5-Omni with +19.05 on DailyOmni (cross-modal understanding), +1.7 on MMAR (audio), and +3.9 on Video-MME (vision), while using just 0.2T training tokens - a 6× reduction compared to Qwen2.5-Omni's 1.2T. We finally demonstrate omni-modal advantages in downstream applications spanning robotics, medical AI, and smart factory.

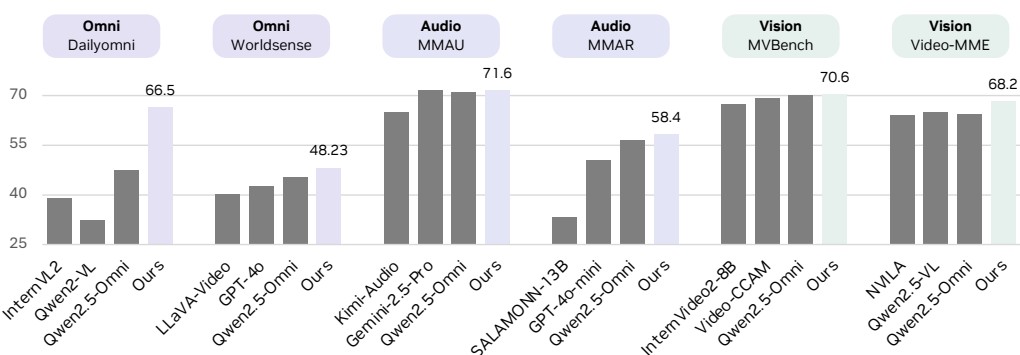

Figure 1: OmniVinci demonstrates strong performance across widely used omni-modal (+19.05 on Dailyomni), audio (+1.7 on MMAR), and vision (+3.9 on Video-MME) understanding benchmarks.

# 1 Introduction

The progress of multimodal LLMs has demonstrated appealing applications when LLMs learn to see with vision (Lin et al., 2024b; Liu et al., 2023; Alayrac et al., 2022) or listen with audio (Goel et al., 2025; Chu et al., 2023; Tang et al., 2023a). Recent work has enabled joint video-audio alignment, further unifying their strengths towards general intelligence (OpenAI, 2024; Wu et al., 2024b; Tang

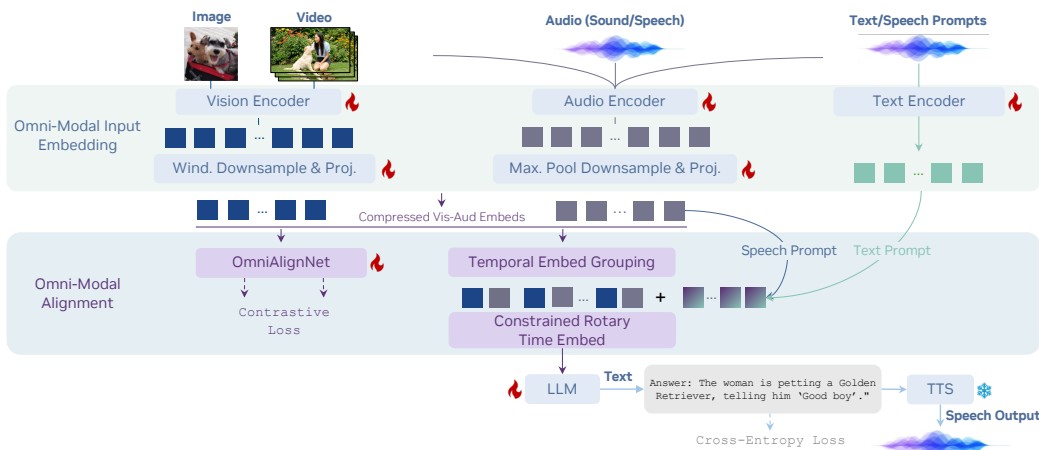

Figure 2: We introduce a foundation model for omni-modal understanding. Our model blends information from vision, audio, and text modalities into a unified omni-modal token sequence via the proposed omni-modal alignment mechanism.

et al., 2023b; Ye et al., 2024; Abouelenin et al., 2025; Xu et al., 2025). However, training such an omni-modal system can be expensive and challenging across many dimensions, as it relies on proper choices of network architecture and data recipe.

This work presents a systematic exploration of developing omni-modal LLMs aiming to enable simultaneous understanding of vision, audio (encompassing both natural sounds and human speech), and language. We ablate and validate the design choices overseeing model architecture design, data curation, and training strategy. For model architecture, we introduce a new framework to harmonize vision and audio embeddings in a unified omni-modal embedding space, featuring three new techniques: (i) *OmniAlignNet* that learns to construct a modality-shared space to align vision and audio embeddings from the same video; (ii) *Temporal Embedding Grouping* that divides the time dimension into multiple chunks and reorganizes the vision and audio embeddings according to their timestamps to align with the corresponding chunks; (iii) *Constrained Rotary Time Embedding* to directly insert periodic temporal information into vision-audio embeddings. We observe noticeable performance improvements with these techniques, as shown later in our experiments. On the data front, we curate 24 million high-quality multimodal conversation samples that span a diverse set of tasks across audio, video, and image domains, including both modal-specific conversations and omni-modal conversations. We tackle the scarcity of omni-modal data by exploiting existing video-with-audio QA data, which implicitly encodes omni-modal signals (*implicit learning*). To further facilitate omni-modal learning, we generate synthetic conversations with explicit omni-modal labels (*explicit learning*).

Our findings enable a frontier omni-modal model, named OmniVinci. See a quick performance comparison in Figure 1 and more in our experimental section. Compared to prior art such as Qwen2.5-Omni and Gemini-2.5-Pro, OmniVinci further pushes the boundary of various multimodal understanding tasks, with gains of +2.83% on WorldSense and +19.05% on Dailyomni for joint vision-audio understanding, +1.7% on MMAR for audio understanding, and +3.9% on Video-MME for vision understanding. OmniVinci also pushes on efficiency fronts using only 0.2T training tokens, around 6× fewer than Qwen2.5-Omni's 1.2T tokens. More encouragingly, we observe the synergy between audio and video, not only for perception, but also for reasoning. Finally, we demonstrate that OmniVinci has enabled or improved a wide range of important downstream applications, including robotics, video broadcasting, medical, and smart factory use cases.

## 2 MODEL ARCHITECTURE

The key objective of model architecture design is to support composable cross-modal understanding through integrating heterogeneous input from images, videos, audio, and text, into a shared omni-modal latent space. As shown in Figure 2, we adopt an auto-regressive regime to encode visual and audio signals, and then align them as input of LLM backbone.

**Omni-Modal Input Embedding.** To simplify the network design, we (i) decompose video into a sequence of temporally correlated images and audio, and (ii) employ a unified audio encoder to

handle both acoustic and speech information in context and prompt. We present the encoder sharing paths in Figure 2, and describe the details of encoding streams in Appendix D.1.

## 2.1 OMNI-MODAL ALIGNMENT MECHANISM

We next integrate embeddings from all modalities into a unified latent space as input for LLM.

**OmniAlignNet module.** For a given input video, the audio and vision streams have an inherent semantic connection, providing complementary information for each other. Such a correlation provides a natural way to more effectively learn and align vision and audio embeddings in the unified latent space. To this end, we propose OmniAlignNet, which strengthens the learning of vision and audio embeddings via exploiting their complementary information. As illustrated in Figure 3, the OmniAlignNet module first maps visual and audio embedding sequences (outputs of modality-specific projectors) into a shared latent embedding space and then aligns them via contrastive learning, inspired by ImageBind (Girdhar et al., 2023).

Given an input video with an accompanying audio stream, we denote the sequence of visual embeddings produced by the visual projection layer as $\mathbf{E}_v \in \mathbb{R}^{N_v \times C}$ and the sequence of audio embeddings produced by the audio projection layer as $\mathbf{E}_a \in \mathbb{R}^{N_a \times C}$, with $N_v$ and $N_a$ represent the number of visual and audio embeddings, respectively, while $C$ denotes the latent dimensionality. To align representations, we initialize a vision query embedding $\mathbf{Q}_v \in \mathbb{R}^{1 \times C}$ and an audio query embedding $\mathbf{Q}_a \in \mathbb{R}^{1 \times C}$. These queries are used to project $\mathbf{E}_v$ and $\mathbf{E}_a$ into fixed-size embeddings of shape $(1 \times C)$. Suppose each batch has $K$ videos, the projected features are then processed through three layers of self-attention modules and L2 normalized, yielding the vision-omni embedding $\mathbf{V} \in \mathbb{R}^{K \times C}$ and the audio-omni embedding $\mathbf{A} \in \mathbb{R}^{K \times C}$, respectively, in a modality-shared latent space.

With embeddings $\mathbf{V}$ and $\mathbf{A}$ in the shared latent space, we now apply CLIP-style contrastive loss (Radford et al., 2021) on the output embeddings to minimize intra-sample cross-modal distance, while maximizing inter-sample cross-modal distance. Let $\{\mathbf{V}_i, \mathbf{A}_i\}_{i=1}^K$ be the set of L2-normalized visual and audio embeddings for a batch of $K$ video clips. The similarity between the $i$-th visual embedding and the $j$-th audio embedding is computed as their dot product, $s_{ij} = \mathbf{V}_i^T \mathbf{A}_j$. The contrastive loss is then formulated as a symmetric cross-entropy loss

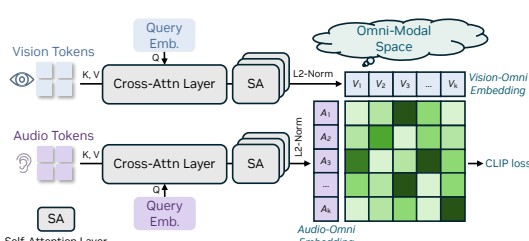

Figure 3: Illustration of the proposed OmniAlignNet module.

over the similarity score. The loss for aligning vision to audio ($L_{v \to a}$) and audio to vision ($L_{a \to v}$) is:

$$L_{v \to a} = -\frac{1}{N}\sum_{i=1}^{N} \log \frac{\exp(s_{ii})}{\sum_{j=1}^{N} \exp(s_{ij})}, L_{a \to v} = -\frac{1}{N}\sum_{i=1}^{N} \log \frac{\exp(s_{ii})}{\sum_{j=1}^{N} \exp(s_{ji})}. \quad (1)$$

The final objective for the OmniAlignNet module, $L_{\text{o-align}}$, is the average of these two directional losses, encouraging a bidirectional alignment between the modalities: $L_{\text{o-align}} = \frac{1}{2}(L_{v \to a} + L_{a \to v})$.

While OmniAlignNet effectively aligns the high-level semantics of visual and audio embeddings, it falls short in modeling their temporal relationships. To overcome this limitation, we introduce two techniques: Temporal Embedding Grouping and Constrained Rotary Time Embedding, which are described in the following sections.

**Temporal Embedding Grouping (TEG).** We first impose temporal order to visual-audio embeddings by organizing them into groups based on their timestamps. The relative temporal order information is then encoded in the position of visual and audio embeddings in the input sequence.

Let the duration of each temporal group be $T_G$, which controls the granularity of the grouping. For simplicity, consider a case where we only sample four visual frames at timestamps $\{t_v^1, t_v^2, t_v^3, t_v^4\}$ and four audio samples at timestamps $\{t_a^1, t_a^2, t_a^3, t_a^4\}$. These timestamps satisfy $t_v^1 < t_v^2 < T_G < t_v^3 < t_v^4 < 2T_G$ and $t_a^1 < t_a^2 < T_G < t_a^3 < t_a^4 < 2T_G$. The corresponding set of visual embeddings is $E_v = \{\mathbf{e}_v^{t_v^1}, \mathbf{e}_v^{t_v^2}, \mathbf{e}_v^{t_v^3}, \mathbf{e}_v^{t_v^4}\}$, where each embedding $\mathbf{e}_v \in \mathbb{R}^{(HW) \times C}$. Here, $H$ and $W$ represent the height and width of the visual feature map, and $C$ is the latent dimension. Similarly, the set of audio

embeddings is $E_a = \{\mathbf{e}_a^{t_a^1}, \mathbf{e}_a^{t_a^2}, \mathbf{e}_a^{t_a^3}, \mathbf{e}_a^{t_a^4}\}$, with each $\mathbf{e}_a \in \mathbb{R}^{1 \times C}$. Based on their timestamps relative to the duration $T_G$, the embeddings for each modality are partitioned into two temporal groups:

$$G_v^1 = \{\mathbf{e}_v^{t_v^1}, \mathbf{e}_v^{t_v^2}\}, G_v^2 = \{\mathbf{e}_v^{t_v^3}, \mathbf{e}_v^{t_v^4}\}, G_a^1 = \{\mathbf{e}_a^{t_a^1}, \mathbf{e}_a^{t_a^2}\}, G_a^2 = \{\mathbf{e}_a^{t_a^3}, \mathbf{e}_a^{t_a^4}\}. \tag{2}$$

Then we combine the visual and audio groups based on temporal order, and obtain the omni-modal embedding sequence:

$$\mathbf{E}_{\text{group}} = \left[G_v^1, G_a^1, G_v^2, G_a^2\right] = \left[\mathbf{e}_v^{t_v^1}, \mathbf{e}_v^{t_v^2}, \mathbf{e}_a^{t_a^1}, \mathbf{e}_a^{t_a^2}, \mathbf{e}_v^{t_v^3}, \mathbf{e}_v^{t_v^4}, \mathbf{e}_a^{t_a^3}, \mathbf{e}_a^{t_a^4}\right]. \tag{3}$$

This temporal organization of the embedding sequence allows the subsequent LLM backbone to better capture the temporal relationships among embeddings from different modalities. Our experiments show that this time-based grouping improves the model's ability to comprehend information from multiple modalities.

**Constrained Rotary Time Embedding (CRTE).** TEG incorporates relative temporal order into embeddings but still lacks the ability to encode absolute timestamp information. Prior work, RoTE (Goel et al., 2024), explored embedding rotations to inject absolute timestamps, but this method remains sensitive to minor timestamp fluctuations and struggles to capture larger temporal shifts effectively. To overcome these limitations, we introduce a constrained timestamp embedding strategy that defines a maximum time horizon, $T_{\max}$, enabling a more balanced temporal sensitivity. Our approach comprises three stages: base frequency construction, frequency modulation, and element-wise rotary embedding, as described next.

**Base Frequency Generation.** We first define base frequencies as:

$$\omega_i = \frac{2\pi}{T_{\max}\theta^{i/C}}, \quad \text{for} \quad i = 0, 1, \dots, C-1, \tag{4}$$

where $\omega_i$ is the base frequency for dimension $i$, $C$ is the embedding dimension, $\theta \geq 1$ controls frequency scaling, and $T_{\max}$ defines the coarsest temporal resolution. A smaller $T_{\max}$ increases frequency and sensitivity to fine-grained differences, while a larger one captures broader trends but may blur close timestamps, and is thus critical for balancing local and global temporal encoding.

**Frequency Modulation.** To adapt frequencies to actual timestamps, we scale them as: $\Omega_{i,j} = \omega_i \cdot t_j$, where $\Omega_{i,j}$ is the modulated frequency at dimension $i$ and time $t_j$ for sample $j$. This step ensures that temporal differences are reflected in the rotation applied to embeddings.

**Rotary Embedding Application.** Similar to RoPE (Su et al., 2024), given an embedding vector $\mathbf{x} \in \mathbb{R}^C$ of sample $j$ (a sampled frame for video or a sampling point for audio), we apply rotation as:

$$\text{CRTE}(\mathbf{x}, \Omega_{:,j}) = \mathbf{x} \odot \cos(\Omega_{:,j}) + \texttt{RotateHalf}(\mathbf{x}) \odot \sin(\Omega_{:,j}), \tag{5}$$

where $\odot$ denotes element-wise multiplication, and $\texttt{RotateHalf}$ rotates each pair of dimensions by $90^0$: $\texttt{RotateHalf}(\mathbf{x}) = [-x_2, x_1, -x_4, x_3, \dots, -x_C, x_{C-1}]$. The $\texttt{RotateHalf}$ function effectively groups the entire $C$-dimensional embedding vector into $C/2$ independent 2D planes. Each of these 2D planes gets its own rotation, and the angle of rotation can be different for each pair. We apply rotations at varying frequencies across different pairs of dimensions for two primary reasons: it enables a rich, multi-scale representation of temporal information, and it preserves the semantic integrity of the original embedding vectors.

**Final Embedding Sequence.** After CRTE, the temporally-aligned omni-modal embedding sequence is passed into the LLM backbone, allowing it to integrate both fine- and coarse-grained timing cues during downstream processing.

**Input-Output Configuration.** The final architecture perceives flexible input modality combinations with a subset or union of all modalities, *e.g.*, video with or without audio, with speech or text prompts. On the output end, the text-output based system can be connected with off-the-shelf Text-to-Speech (TTS) modules – we analyze their tradeoffs in Section E.4. Without bells and whistles, users can generate spoken descriptions for videos, answer spoken questions, or verbally instruct robots.

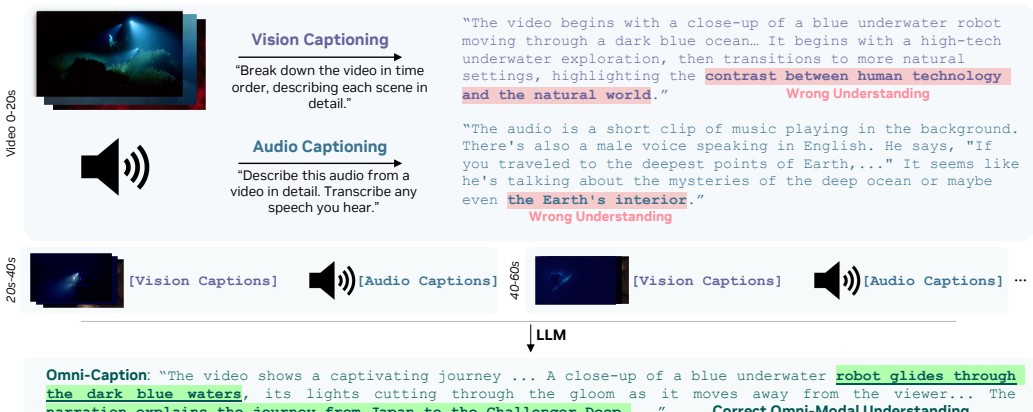

Figure 4: Omni-modal captions generation pipeline. Video is segmented into 20-second clips. Visual and audio captions are generated independently for each segment, but lack cross-modal context and contain wrong understanding (modality-specific hallucination). A separate LLM performs cross-modal correction and summarization to create accurate omni-modal captions.

## 3 TRAINING STRATEGY

To gradually enable comprehensive omni-modal understanding of a pretrained LLM, we use a two-stage approach: we first conduct modality-specific training to develop individual capabilities for each modality, followed by omni-modal joint training to integrate these capabilities.

### 3.1 MODALITY-SPECIFIC TRAINING

We use a two-stage approach for training: Starting from a pretrained LLM, we first conduct modality-specific training to develop individual capabilities for each modality, followed by omni-modal joint training to integrate these capabilities. Due to space limitations, we present comprehensive details of this phase in Appendix D.3 and proceed directly to describe the subsequent omni-modal joint training phase in the next section.

### 3.2 OMNI-MODAL JOINT TRAINING

We employ two types of data in the omni-modal joint training phase: (i) modality-specific data, randomly sampled from the datasets used in the earlier vision-only and audio-only training, and (ii) omni-modal data, which contains both vision and audio inputs. For the omni-modal data, which contains both visual and audio inputs, can be further divided into two categories, *i.e.*, implicit omni-modal learning data and explicit omni-modal learning data, depending on how the omni-modal understanding ability is supervised in training.

**(i) Implicit Learning Data.** Videos are naturally omni-modal when visual and audio streams are present simultaneously but remains under explored. We first take advantage of the existing video QA datasets to supervise the visual-audio joint understanding ability implicitly, which is underutilized in most previous video LLMs. This practice, we refer as *implicit omni-modal learning*, leads to notably improved performance in video understanding that remains under utilized by prior work.

**(ii) Explicit Learning Data.** To obtain more direct and accurate supervision for joint visual-audio understanding ability, we further propose an omni-modal data engine to synthesize omni-modal labeling for videos with audio tracks, enabling us to conduct *explicit omni-modal learning*.

**Omni-Modal Data Engine.** The whole data engine is visualized in Figure 4. We start with synthetic audio and video captions using pretrained vision captioning model (Zhu et al., 2025) and audio captioning model (Xu et al., 2025). We immediately observed that captions generated from either modality alone can lead to wrong understanding due to the inherent modality-specific limitations. As illustrated in Figure 4, the video is centered around deep-sea exploration. However, the vision-captioning model incorrectly interpreted it as being only about human technology, relying solely on visual cues without access to the speech in video. Conversely, the audio-captioning model wrongly labeled it as related to "Earth's interior", since it could only draw meaning from the audio track. We refer to this limitation as "*modality-specific hallucination*". To address this issue, we employ a LLM (Yang et al., 2025a) to correct and summarize the visual and audio captions based on information

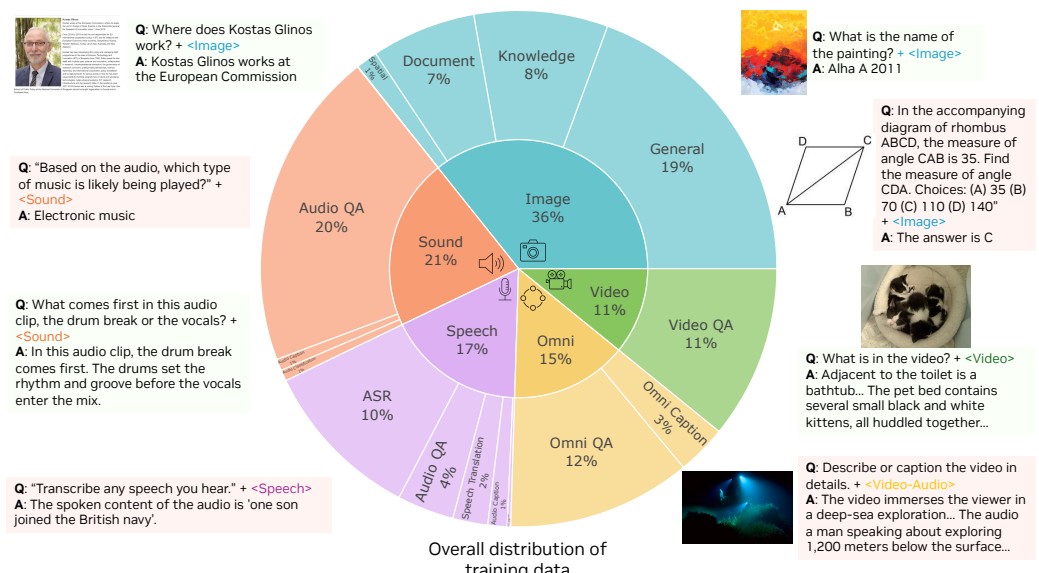

Figure 5: Pie chart of overall distribution of training data across modalities, showing proportions for image (36%), non-speech sound (21%), speech (17%), omni (15%), and video (11%).

from both sides, producing a comprehensive joint caption for each 2-minute segment. From our observation, this method can help achieve correct omni-modal understanding, as shown in the example in Figure 4. Furthermore, we enhance the diversity and quality of the omni-modal data by synthesizing QA pairs with reasoning trace from the omni-modal captions using a reasoning LLM (Guo et al., 2025a). The resulting dataset greatly assists with learning, as we show in experiments.

> **Key Insight 1.** Captioning based solely on audio or visual is often inaccurate because of the inherent limitations of each modality. Hence, a joint captioning approach is preferred to integrate both modalities and produce comprehensive summaries across clips.

**Joint Training Data Distribution.** As shown in Figure 5, the overall training dataset contains 24 million modality-specific conversations from 150+ sub-datasets across image, video, and audio understanding tasks. Omni-modal data contributes 15%, image data constitutes the largest share at 36%, speech data represents 17% of the total, and video data forms the remaining 11%. For more details, please refer to Appendix D.5. To enable audio-prompted ability, we convert text prompts in multimodal tasks into speech using Magpie TTS model (Hussain et al., 2025; Neekhara et al., 2024; Casanova et al., 2025), generating omni-modal speech-visual input pairs. The questions are generated from a comprehensive collection of multimodal datasets, including general multimodal understanding, image captioning, spatial relationship reasoning and referring, chart and table interpretation, scientific figure analysis, document understanding, and multi-hop reasoning. This diverse range enables comprehensive evaluation across core vision-language capabilities such as factual grounding, reasoning over structured data, and complex multi-step inference in both scientific and general domains. See detailed distribution of speech-prompted omni QA datasets in Figure 14.

# 4 EXPERIMENTS

We start with ablations to validate our design options in Section 4.1, before large-scale training towards frontier performances in Section 4.2.

## 4.1 DESIGN CHOICE ABLATION

### 4.1.1 VISUAL-AUDIO ALIGNMENT SCHEME

**Baseline Setup.** To investigate the behavior of omni-modal models under

Table 1: Ablation study for omni-modal alignment. The proposed Temporal Embedding Grouping (TEG), Constrained Rotary Time Embedding (CRTE), and OmniAlignNet consistently achieve better average performance across modalities.

| Method | Omni | | | |
|---|---|---|---|---|
| | Worldsense ↑ | Dailyomni ↑ | Omnibench ↑ | Average ↑ |
| Token Concatenation – Baseline | 42.21 | 54.55 | 36.46 | 45.51 |
| + TEG (**ours**) | $44.51_{+2.30}$ | $60.99_{+6.44}$ | $37.65_{+1.19}$ | $47.72_{+2.21}$ |
| ++ Learned Time Embedding | $44.58_{+2.37}$ | $60.40_{+5.85}$ | $36.91_{+0.45}$ | $47.30_{+1.79}$ |
| ++ RoTE | $44.42_{+2.21}$ | $60.74_{+6.19}$ | $38.24_{+1.78}$ | $47.80_{+2.29}$ |
| ++ CRTE (**ours**) | $45.46_{+3.25}$ | $65.66_{+11.11}$ | $39.64_{+3.18}$ | $50.25_{+4.74}$ |
| +++ OmniAlignNet (**ours**) | $\mathbf{46.21}_{+4.00}$ | $\mathbf{65.83}_{+12.28}$ | $\mathbf{45.74}_{+9.28}$ | $\mathbf{52.59}_{+7.08}$ |

various experimental conditions we
gradually introduce new techniques onto a baseline model trained with 10B tokens randomly sampled
subset of the full data mixture (the sampling process is weighted according to the original dataset
sizes). We evaluate model performance on Worldsense (Benchekroun et al., 2023), Dailyomni (Zhou
et al., 2025), and Omnibench (Li et al., 2024c).

**Temporal Embedding Grouping.** We observe immediate performance improvements with TEG
applied to the baseline and present the results in Table 1, thanks to the enhanced temporal alignment
of modality tokens.

**Constrained Rotary Time Embedding.** We next compare CRTE with other design choices: *(i)*
*"Learned Time Embedding"* that defines a trainable embedding matrix, where each discrete timestamp
in the range $[0, T_{max}]$ is mapped to a unique vector via MLP. *(ii) "RoTE" (Goel et al., 2024)*, a
recent embedding method introduced in Section 2.1. As summarized in Table 1, the "Learned
Time Embedding" method slightly degrades performance (47.30), indicating it is unsuitable for
absolute timestamps. RoTE offers only marginal gains, while the proposed Constrained Rotary Time
Embedding achieves the best score (50.25), clearly improving over the baseline.

**OmniAlignNet.** Finally, we impose the proposed OmniAlignNet on top of TEG and CRTE. As
shown in the bottom section of Table 1, OmniAlignNet delivers significant performance boosts across
all benchmarks. The average score improves from 50.25 to 52.59 (+2.34), and the model achieves
considerable gains on Omnibench (+6.1), Worldsense (+0.75), and Dailyomni (+1.17).

### 4.1.2 IMPLICIT AND EXPLICIT LEARNING

We next validate implicit and explicit
omni-modal learning as detailed in
Section 3.2. For implicit learning, we
continue to finetune the above model
on 270K video conversations with au-
dio stream. Results in Table 2 show
clear gains on VideoMME (Fu et al.,
2024a), even when subtitles are pro-
vided, highlighting the value of learn-

Table 2: Ablation study on joint visual-audio learning meth-
ods. "Visual+Audio" uses audio in video for implicit learning
(IL), while "data engine" generates omni-modal data for ex-
plicit learning (EL).

| Method | VideoMME ↑ | | VideoMME w/o sub. ↑ | | |
|---|---|---|---|---|---|
| | w/ subtitles | w/o subtitles | Short | Medium | Long |
| Visual Alone | 66.37 | 61.67 | 74.22 | 59.67 | 51.11 |
| Visual + Audio (IL) | 66.96$_{+0.59}$ | 63.76$_{+2.09}$ | 71.31$_{-2.91}$ | 64.16$_{+4.49}$ | 55.82$_{+4.71}$ |
| Visual + Audio + Data Engine (EL) | 68.63$_{+2.26}$ | 67.37$_{+5.70}$ | 76.78$_{+2.56}$ | 67.56$_{+7.89}$ | 57.78$_{+6.67}$ |

ing directly from audio. Further adding explicit learning data from our omni-modal data engine yields
stronger improvements across benchmarks, showing the effectiveness of our data pipeline.

## 4.2 SCALING AND EVALUATION

With validated design choices, we now scale up the experiments using the full post-training omni-
modal dataset introduced in Section 3.2. Training details are in Appendix D.4.

### 4.2.1 OMNI-MODAL BENCHMARK

We first evaluate on omni-modal understand-
ing benchmarks and show results in Table 21.
OmniVinci sets a new state-of-the-art average
score of 53.73, and marks a notable improve-
ment of **+4.07** compared to the next best model,
Qwen2.5-Omni. On the Worldsense benchmark,
our model achieves the highest score of 48.23,
surpassing Qwen2.5-Omni by **+2.83**. The ad-
vantage is even more significant on the Daily-
omni dataset, where our model attains a score of
66.50, leading by **+19.05** over Qwen2.5-Omni
and by **+5.18** over Gemini-2.0-Flash-Lite. In the

Table 3: Omni benchmarks, including video–audio
datasets Worldsense and Dailyomni, as well as the
image–audio dataset Omnibench.

| Model | Omni | | | |
|---|---|---|---|---|
| | Worldsense
(*Video–Audio* ↑) | Dailyomni
(*Video–Audio* ↑) | Omnibench
(*Image–Audio* ↑) | Avg.
(↑) |
| Gemini | – | 61.32 (2.0 Flash Lite) | 42.91 (1.5 Pro) | - |
| GPT-4o | 42.60 | – | – | - |
| InternVL2 | 39.10 | – | 47.55 (v2.5) | - |
| Qwen2-VL | 32.40 | – | 48.60 | - |
| Qwen2.5-Omni | 45.40 | 47.45 | **56.13** | 49.66 |
| **OmniVinci** | **48.23** | **66.50** | 46.47 | **53.73** |

Omnibench benchmark, our model shows a solid score of 46.47, higher than Gemini 1.5 Pro.

### 4.2.2 AUDIO BENCHMARK

**Audio QA.** We assess our model on audio understanding benchmarks, MMAR (Ma et al., 2025) and

Table 5: Video benchmarks. OmniVinci outperforms NVILA baseline.

| Model | | LongVideoBench ↑ | | MVBench ↑ | Video-MME ↑ |
|---|---|---|---|---|---|
| | | val | test | test | w/o sub. |
| GPT-4o mini | - | 56.5 | 58.8 | – | 64.8 |
| GPT-4o | - | 66.7 | 66.7 | – | 71.9 |
| LLaVA-NeXT-Video | 7B | 43.5 | 43.5 | 33.7 | 46.5 |
| InternVL2 | 8B | 54.6 | – | 65.8 | 56.3 |
| LLaVA-OneVision | 8B | 56.5 | – | 56.7 | 58.2 |
| LongVILA | 7B | 57.1 | – | 67.1 | 60.1 |
| Qwen2.5-VL | 8B | 56.0 | - | 69.6 | 65.1 |
| InternVL3 | 8B | 58.8 | – | 75.4 | 66.3 |
| Qwen3-VL | 8B | – | – | 68.7 | 71.4 |
| Qwen2.5-Omni | 11B | - | – | 70.3 | 64.3 |
| NVILA | 8B | 57.7 | 58.7 | 68.1 | 64.2 |
| **OmniVinci** | 9B | 61.3 | 62.0 | 70.6 | 68.2 |

Table 6: Image benchmarks. OmniVinci maintains comparable image understanding performance with NVILA.

| Model | | AI2D test | ChartQA test | DocVQA test | InfoVQA test | MathVista testmini | MMMU | | | Real-WorldQA | SEED image | TextVQA val | VQAv2 testdev |
|---|---|---|---|---|---|---|---|---|---|---|---|---|---|
| | | | | | | | val | test | pro | | | | |
| GPT-4o | – | 94.2 | 85.7 | 92.8 | 79.2 | 63.8 | 69.1 | 64.7 | 51.9 | 75.4 | 76.2 | 77.4 | 78.7 |
| Claude 3.5 Sonnet | – | 94.7 | 90.8 | 85.2 | 74.3 | 67.7 | 68.3 | 63.7 | 51.5 | 60.1 | – | 74.1 | 70.7 |
| Gemini 1.5 Pro | – | 94.4 | 87.2 | 93.1 | 81.0 | 63.9 | 62.2 | 57.6 | 43.5 | 70.4 | – | 78.7 | 80.2 |
| LLaVA-1.5 | 7B | 55.5 | 17.8 | 28.1 | 25.8 | 25.6 | 35.7 | – | – | 54.8 | 66.1 | 58.2 | 78.5 |
| VILA-1.5 | 8B | 76.6 | 52.7 | 40.6 | 25.9 | 36.7 | 38.6 | 32.7 | – | 52.7 | 73.8 | 68.5 | 83.0 |
| Cambrian-1 | 8B | 73.0 | 73.3 | 77.8 | 41.6 | 49.0 | 42.7 | – | – | 64.2 | 74.7 | 71.7 | 81.2 |
| Florence-VL | 8B | 74.2 | 74.7 | 84.9 | 51.7 | 55.5 | 43.7 | – | – | 64.2 | 74.9 | 74.2 | 84.7 |
| LLaVA-OneVision | 8B | 81.4 | 80.0 | 87.5 | 68.8 | 63.2 | 48.8 | 42.8 | 24.1 | 66.3 | 75.4 | 78.3 | 84.0 |
| Llama 3.2 | 11B | 91.9 | 83.4 | 88.4 | – | – | 50.7 | – | – | – | – | – | 75.2 |
| InternVL2 | 8B | 83.8 | 83.3 | 91.6 | 74.8 | 58.3 | 51.2 | 42.6 | 29.0 | 64.2 | 76.2 | 77.4 | 76.7 |
| Qwen2-VL | 8B | 83.0 | 83.0 | 94.5 | 76.5 | 58.2 | 54.1 | 46.6 | 30.5 | 70.1 | 76.0 | 84.3 | 82.9 |
| InternVL3 | 8B | 85.2 | 86.6 | 92.7 | 76.8 | 75.2 | 53.3 | 65.6 | – | 70.8 | 76.2 | 80.2 | – |
| Qwen3-VL | 8B | 85.7 | – | 96.1 | 83.1 | 77.2 | 69.6 | – | 55.9 | 71.5 | – | – | – |
| NVILA | 8B | 92.3 | 86.1 | 93.7 | 70.7 | 65.4 | 49.9 | 44.4 | 27.8 | 68.6 | 76.5 | 80.1 | 85.4 |
| **OmniVinci** | 9B | 91.5 | 84.6 | 91.5 | 69.0 | 63.5 | 49.7 | 44.6 | 26.4 | 67.5 | 77.1 | 83.9 | 85.4 |

MMAU (Sakshi et al., 2024), with results reported in Tables 4 and 16. On MMAR, OmniVinci surpasses Qwen2.5-Omni by +1.7, and on MMAU by +0.6, highlighting significant improvement in general audio understanding.

Table 4: Audio QA benchmark.

| Model | MMAR (↑) |
|---|---|
| LTU | 19.20 |
| Audio Flamingo 2 | 21.90 |
| Qwen-2-Audio | 30.40 |
| SALAMONN | 33.20 |
| Baichuan-Omni-1.5 | 40.70 |
| Qwen2.5-Omni | 56.70 |
| **OmniVinci** | **58.40** |

**Speech Recognition.** To assess the automatic speech recognition (ASR) capabilities of OmniVinci, we evaluate it on four widely used benchmarks: LibriSpeech (Panayotov et al., 2015), AMI (Kraaij et al., 2005), Tedlium (Rousseau et al., 2012), and VoxPopuli(Wang et al., 2021), comparing against leading multi-modal models. As shown in Table 7, our model achieves competitive word error rates (WER) of 1.7 on LibriSpeech-clean and 3.7 on LibriSpeech-other, closely matching or surpassing the latest works.

We further investigate OmniVinci's performance under two agentic-cascaded setups: (i) incorporating ASR text history (Huang et al., 2025) and (ii) leveraging retriever-based training as shown in Figure 16. These techniques help boost OmniVinci's capacity, yielding average WERs of **5.7** and **5.0**, respectively. These test-time scaling studies are provided in Appendix E.3 (Table 19).

Table 7: Multi-domain speech recognition benchmarks. *Results taken from related papers; details in Appendix E.3.

| Model | WER (↓) | | | | | |
|---|---|---|---|---|---|---|
| | LS$_{clean}$ | LS$_{other}$ | AMI | Ted. | Vox. | Avg. |
| Whisper-large-v3 | 1.8 | 3.6 | 16.1 | 3.9 | 10.1 | 7.1 |
| Qwen2-Audio | 1.7 | 4.1 | 15.2 | 3.1 | 7.1 | 6.4 |
| GPT-4o-real-time | 2.5 | 5.0 | 19.3 | 4.1 | 12.1 | 8.6 |
| Gemini-2.0-Flash | 2.5 | 5.9 | 21.5 | 3.0 | 7.9 | 8.2 |
| Phi-4-MM | **1.7** | 3.8 | **11.5** | **2.9** | 5.9 | **5.2** |
| Qwen2.5-omni | 1.8* | **3.4*** | 17.9 | 5.2 | **5.8*** | 6.8 |
| **OmniVinci** | 1.7 | 3.7 | 16.1 | 3.4 | 6.8 | 6.3 |

### 4.2.3 VIDEO BENCHMARK

We compare with other open-source video-language models in Table 5. On the LongVideoBench (Wu et al., 2024a) val set, OmniVinci achieves a score of 61.3, outperforming NVILA by a margin of **+3.6**. Similarly, our model improves on MVBench (Li et al., 2024b) with a score of 70.6, outperforming also the recently released Qwen2.5-Omni (70.3).

Furthermore, on the Video-MME (Fu et al., 2024a) benchmark (without subtitles hints), OmniVinci again sets a high score at 68.2, surpassing Qwen2.5-VL-7B by **+3.1**, positioning it as a leading open-source model for video comprehension tasks.

> **Key Insight 2.** Audio understanding capacity enables consistent metric improvements across video benchmarks, akin to human perception.

### 4.2.4 IMAGE BENCHMARK

We evaluate OmniVinci on ten image benchmarks to test its versatility. These tasks range from understanding diagrams and charts (AI2D (Kembhavi et al., 2016), ChartQA (Masry et al., 2022)) to document analysis (DocVQA (Mathew et al., 2021)), mathematics (MathVista (Lu et al., 2024b)) and general visual question answering (VQAv2-testdev (Goyal et al., 2017)). As shown in Table 6, OmniVinci consistently achieves competitive scores across the board.

### 4.3 OMNI-MODAL REASONING

Building on advances in the Group Relative Policy Optimization (GRPO) (Shao et al., 2024) algorithm and prior work on multi-modal reasoning training (Chen et al., 2025; Feng et al., 2025), we next tackle omni-modal reasoning through accommodating audio tokens in addition to visual ones.

Specifically, for each given question and omni-modal input $q = \{q_t, q_v, q_a\}$ ($q_t$ is textual input, $q_v$ is visual input, and $q_a$ is audio input, respectively), the sampling number is $G$, the policy model, under

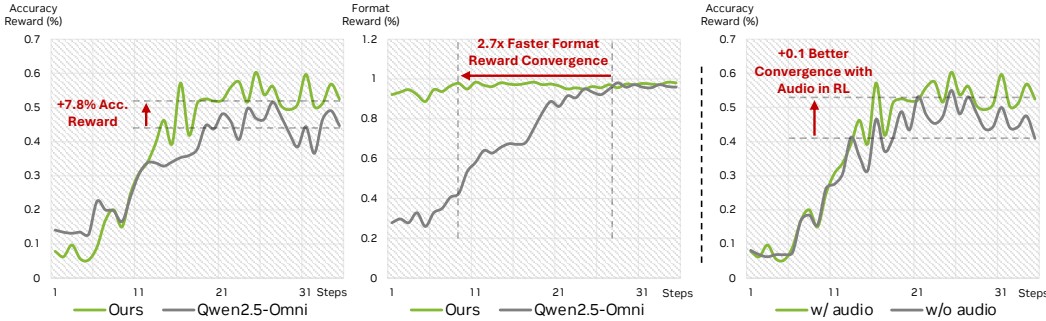

Figure 6: **Left:** Accuracy reward and format reward curves of OmniVinci and Qwen2.5-Omni in RL training. **Right:** Accuracy reward curve of OmniVinci with and without audio.

the old policy $\pi_{\theta_{old}}$, generates a set of candidate answers $\{o_1, o_2, ..., o_G\}$ along with corresponding rewards $\{r_1, r_2, ..., r_G\}$, where the rewards are computed by a rule-based function that evaluates format and accuracy (Shao et al., 2024). The model $\pi_\theta$ is then optimized by maximizing the following objective:

$$\mathcal{J}(\theta) = \mathbb{E}_{q, \{o_i\}} \Big[ \frac{1}{G} \sum_{i=1}^{G} \Big( \min\Big( \frac{\pi_\theta(o_i|q_t, q_v, q_a)}{\pi_{\theta_{old}}(o_i|q_t, q_v, q_a)} A_i, \text{clip}\Big( \frac{\pi_\theta(o_i|q_t, q_v, q_a)}{\pi_{\theta_{old}}(o_i|q_t, q_v, q_a)}, 1 - \epsilon, 1 + \epsilon \Big) A_i \Big)$$
$$- \beta \mathbb{D}_{KL}(\pi_\theta || \pi_{ref}) \Big) \Big], \quad (6)$$

where $\epsilon$ and $\beta$ are hyper-parameters of each loss part, the sampling number $G$ is set as 8. The rewards $\{r_1, r_2, ..., r_G\}$ are normalized to get the advantages ($A_i$) for updating the model:

$$A_i = \frac{r_i - \text{mean}(\{r_1, r_2, ..., r_G\})}{\text{std}(\{r_1, r_2, ..., r_G\})}. \quad (7)$$

We apply GRPO post-training to the final OmniVinci checkpoint to enhance its performance on omni-modal understanding benchmarks. For training data, we curated a 18K omni-modal MCQ dataset using the omni-modal data engine, as detailed in the meth-

Table 8: Ablation study of GRPO post-training.

| Model | Omni | | | |
|---|---|---|---|---|
| | Worldsense (↑) | Dailyomni (↑) | Omnibench (↑) | Avg. (↑) |
| OmniVinci | 48.23 | 66.50 | 46.47 | 53.73 |
| OmniVinci + RL | **48.70**$_{+0.47}$ | **67.08**$_{+0.58}$ | **47.79**$_{+1.32}$ | **54.52**$_{+0.79}$ |

ods section. During GRPO training, we utilize the Long-RL (Chen et al., 2025) as the training framework, configure the model to process up to 64 video frames, with a maximum prompt length of 1024 tokens and a maximum response length of 2048 tokens. The update batch size is set to 64, with the rollout number of 8 for each sample, ensuring robust gradient estimation. We employ a temperature of 1.0 and a top-p value of 0.99 for sampling, facilitating diverse exploration during training. These training configurations are carefully designed to optimize the model's ability to handle complex omni-modal reasoning tasks effectively and efficiently.

As shown in Table 8, we observe consistent performance gains across all benchmarks after applying RL training. Comparing convergence with Qwen2.5-Omni under the same recipe (Figure 6), both models benefit from our multi-modal RL framework, but OmniVinci leverages stronger base performance and instruction-following to surpass Qwen2.5-Omni on the GRPO accuracy curve within 15 steps, while also converging faster on formatting tasks. Ablation experiments further show that including audio input boosts RL effectiveness: with audio, accuracy reward converges +0.1 higher than video-only training (Figure 6, right), highlighting the importance of audio for video learning.

> **Key Insight 3.** Joint audio-visual input surpasses the visual-alone input for GRPO training, offering faster and better convergence.

### 4.4 DOWNSTREAM TASKS

OmniVinci also improves downstream tasks that benefit from video-audio perception, including speech prompted robot navigation (Appendix Sec. C.1), sports video understanding (Appendix Sec. C.2), cross-lingual speech translation (Appendix Sec. C.3), medical analysis considering physician verbal explanations (Appendix Sec. C.4), and semiconductor factory monitoring (Appendix Sec. C.5.1). OmniVinci enables new frontier performances in these domains.

## 5 CONCLUSION

We present OmniVinci, a systematic effort to build an omni-modal LLM that allows joint perception of images, videos, audio, and text. We discuss architectural innovations including OmniAlignNet, Temporal Embedding Grouping, and Constrained Rotary Time Embedding, joint with an enhanced data and training recipe. OmniVinci showcases frontier omni-modal performances, cuts down on training costs, and improves downstream agentic applications.

## ETHICS STATEMENT

This proposed approach exclusively utilizes existing open-source datasets that have been established in previous academic research, without generating or incorporating any novel image, video, or audio materials. The data employed in this study is designated solely for research purposes.

## REPRODUCIBILITY STATEMENT

To facilitate replication by the research community, this project will be publicly available as open-source software. Our model architecture is thoroughly detailed in Section 2, while Section 4 provides comprehensive training procedures and implementation specifics, including hyperparameter configurations.

## LLM USAGE

The authors used LLMs for minor refinement of the wording during writing.

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

APPENDIX

APPENDIX TABLE OF CONTENTS

# A    RELATED WORKS

A significant body of work has focused on augmenting LLMs with individual sensory capabilities, primarily vision and audio, often following a similar architectural blueprint. In the visual domain, the dominant paradigm involves using a vision encoder (*e.g.*, ViT (Dosovitskiy, 2020)) to extract features which are then aligned with the LLM's input space via a bridging module. Pioneering models like Flamingo (Alayrac et al., 2022) introduced sophisticated cross-attention mechanisms, while subsequent works (Li et al., 2023; Zhu et al., 2023; Ye et al., 2023; Driess et al., 2023; Liu et al., 2023; Lin et al., 2024b; Liu et al., 2025a; McKinzie et al., 2024; Dai et al., 2023; Zhang et al., 2023; Wang et al., 2024c; Maaz et al., 2024; Fang et al., 2024; Shi et al., 2025), demonstrated the remarkable effectiveness of a simple projection layer combined with visual instruction tuning. A parallel line of research has applied this pattern to the auditory domain, where Audio-Language Models like LTU (Gong et al., 2023), Whispering-LLaMA (Radhakrishnan et al., 2023), Audio-Flamingo (Goel et al., 2025), Qwen-Audio (Chu et al., 2023), and others (Tang et al., 2023a; Deshmukh et al., 2023; Kong et al., 2024; Ghosh et al., 2025; Huang et al., 2024; Chu et al., 2024) use audio encoders to process speech, music, and ambient sounds. These specialized models represent crucial stepping stones toward the more holistic goal of unified, omni-modal understanding.While specialized models for vision and audio have become increasingly capable, the development of foundational, omni-modal LLMs remains relatively nascent. For example, such a single omni model that can natively process and reason across text, vision, audio, and potentially other data types.

The endeavor presents various challenges in terms of model architecture, data curation, and the immense computational resources required for training. Recent pioneering efforts have addressed the challenges of multimodal understanding and reasoning. Google's Gemini (Google, 2023) represents a significant advancement as a natively multimodal model designed to seamlessly integrate and reason across interleaved text, images, audio, and video inputs. However, it remains proprietary and is not available to the open-source community. Within the open-source community, several noteworthy efforts on omni-modal LLMs have been introduced (Li et al., 2025b; Lu et al., 2024a; Wu et al., 2024b; Ye et al., 2024; Chen et al., 2023c; Hu et al., 2025; Chen et al., 2023b; Liu et al., 2025b; Fu et al., 2024b), demonstrating strong capabilities in joint vision–audio understanding tasks. Among these, Phi-4-MM (Abouelenin et al., 2025) and Qwen2.5-Omni (Xu et al., 2025) achieve the strongest results to date; however, their accompanying technical reports reveal relatively simple architectural choices and a lack of thorough ablation studies to systematically examine critical design decisions. In contrast, our work not only proposes several novel techniques for omni-modal understanding but also adopts a more rigorous experimental approach by conducting comprehensive ablation studies before scaling to large-scale datasets. We systematically evaluate various architectural choices and design decisions, providing detailed experimental analyses that we make publicly available. Through this methodical investigation, we aim to contribute valuable insights that can inform and inspire future research directions in omni-modal large language models.

Some research work study multimodal alignment in other domains. For instance, Cheng et al. (2025) investigates vision–audio alignment for audio generation rather than multimodal understanding as we do. Their method applies RoPE within transformer blocks to align visual and audio tokens, but RoPE encodes only relative temporal information. In contrast, OmniVinci incorporates both relative and absolute temporal cues directly into the input sequence before tokens enter the LLM backbone. Kim et al. (2023) target vision–text alignment and do not address the vision–audio setting. It trains a neural network to predict a text transcription for each frame. By comparison, our TEG and CRTE methods require no training or supervision; instead, they inject temporal information directly using a predefined formula. Guo et al. (2025b) fuse visual and audio tokens at the same timestamp via cross-attention and then forward them into the LLM. However, it does not embed absolute temporal information into the tokens. Our method differs by encoding absolute temporal cues through embedding rotation in CRTE. Finally, there are also research works on time-series series alignment (Zerveas et al., 2021; Eldele et al., 2024) that do not involve vision or audio modalities.

**Compared Models in Experiments.** In the experimental section, we compare with prior works on vision LLMs, audio LLMs, and omni-modal LLMs on various multimodal benchmarks. Specifically, we list the reference here due to the space limit in main text. Compared models include Gemini (Google, 2023; 2024), GPT-4o (OpenAI, 2024), Claude-3.0 (Anthropic, 2024), InternVL2 (Chen et al., 2024c), Qwen2-VL (Wang et al., 2024b), Qwen2.5-Omni (Xu et al., 2025), Phi-4-MM (Abouelenin et al., 2025), Kimi-Audio (KimiTeam et al., 2025), Audio Flamingo 2 (Goel et al., 2025), Qwen2

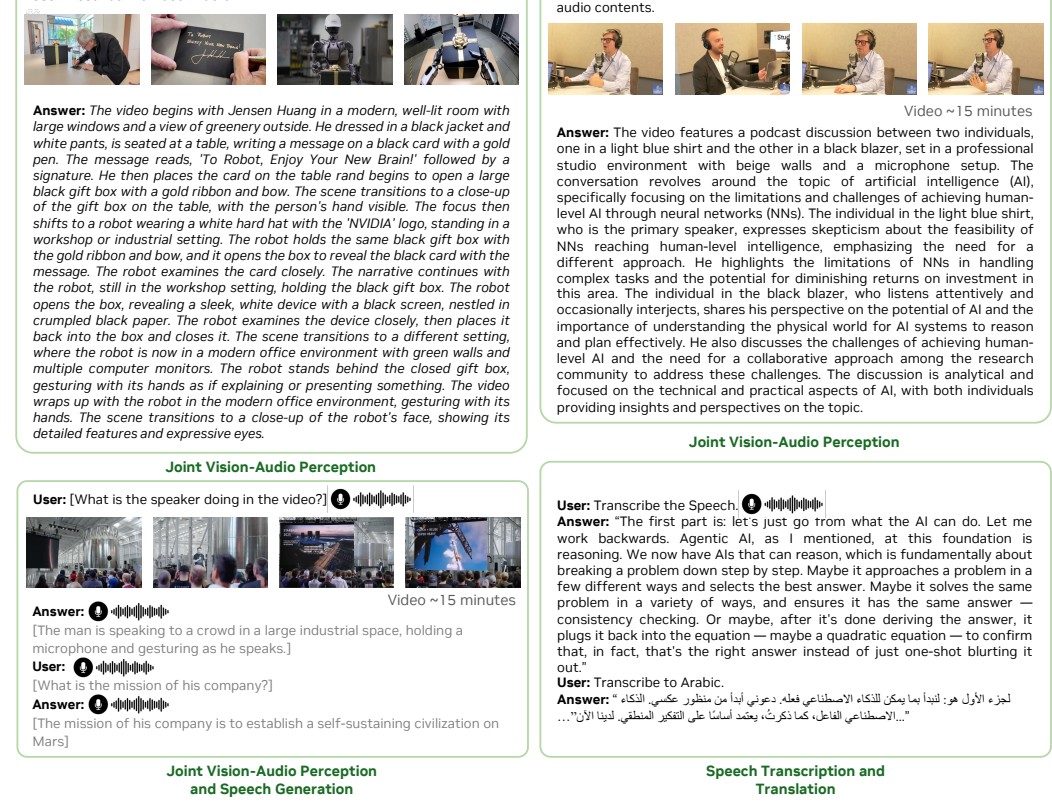

Figure 7: OmniVinci demonstrates strong vision and audio perception capabilities to handle single or joint modality scenarios. The model also supports audio prompts and outputs.

Audio (Chu et al., 2023), Gemma (Team et al., 2024), LTU (Gong et al., 2023), SALAMONN (Tang et al., 2023a), Baichuan-Omni-1.5 (Li et al., 2025b), Whisper-large-3 (Radford et al., 2023), LLaVA-NeXT-Video (Zhang et al., 2024c), InternVL2 (Chen et al., 2024c), LLaVA-OneVision (Li et al., 2024a), LongVILA (Chen et al., 2024a), Qwen2.5-VL (Wang et al., 2024b), NVILA (Liu et al., 2025a), Video-ChatGPT (Maaz et al., 2024), VideoChat2 (Li et al., 2024b).

## B    REAL-WORLD QUALITATIVE STUDY

To evaluate the performance of the model on real-world omni-modal signals, we test it using recently released online videos, as shown in Figure 7. Our results demonstrate that the model can thoroughly comprehend both visual and audio inputs from previously unseen videos and generate responses based on this information, highlighting its strong generalization capabilities. The model successfully integrates speech cues with visual data, allowing for more effective interaction with the environment. These qualitative observations demonstrate the effectiveness of the proposed explicit and implicit training strategy.

## C    DOWNSTREAM AGENTS

Next, we demonstrate the applicability of OmniVinci in a wide range of downstream agentic tasks that yield consistent improvements across benchmarks while enabling new capabilities.

### C.1    ROBOTICS: SPEECH-DRIVEN VISION LANGUAGE NAVIGATION

Prior work (Cheng et al., 2024a; Zhang et al., 2024b; Chen et al., 2023a) in Vision-Language Navigation (Anderson et al., 2018) has predominantly relied on text-based prompts. However, this is not always practical for real-world scenarios where the most convenient and natural way to command a robot is through human speech. As a first step toward this goal, we introduce a speech-driven vision

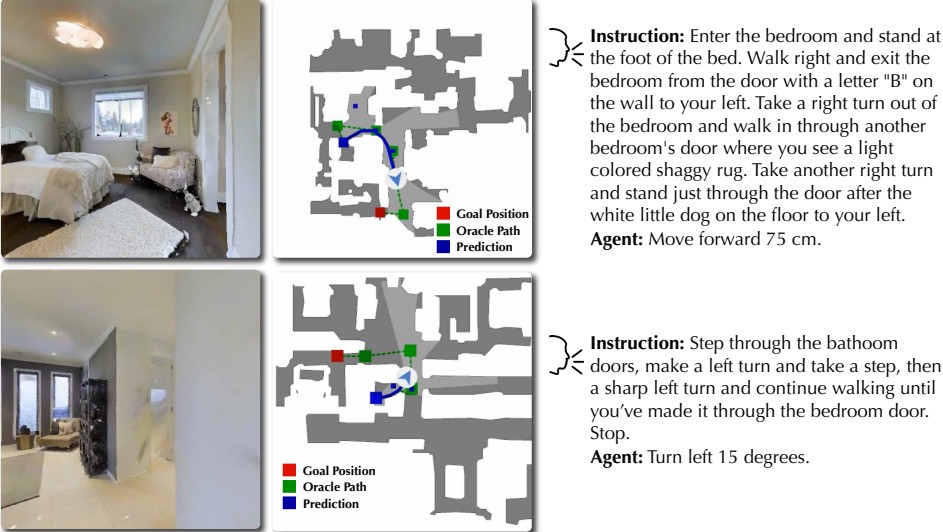

Figure 8: An illustration of our speech-driven navigation agent based on OmniVinci. **Left:** Agent's current visual observation. **Middle:** Top-down map indicating the goal position and the agent's past trajectory. **Right:** the input speech instruction and the agent's predicted action given the current observation.

language navigation task. This task is inherently more challenging than its text-based counterpart, as interpreting the nuances of speech is more complex than processing clean text.

Table 9: Vision Language navigation results on R2R-CE. Our speech-driven model, OmniVinci, achieves comparable performance to the text-driven NVILA, with a lower navigation error.

| Model | Size | Obs. | Instruction | R2R Val-Unseen | | | |
|---|---|---|---|---|---|---|---|
| | | | | NE ↓ | OS ↑ | SR ↑ | SPL ↑ |
| Seq2Seq | – | RGB | Text | 10.10 | 8.0 | 0.0 | 0.0 |
| CMA | – | RGB | Text | 9.55 | 10.0 | 5.0 | 4.0 |
| NaVid | 7B | RGB | Text | 5.47 | 49.0 | 37.0 | 35.0 |
| NVILA | 8B | RGB | Text | **5.43** | 60.4 | **53.3** | **48.8** |
| OmniVinci | 9B | RGB | Audio and/or Text | 5.67 | **60.8** | 50.6 | 45.1 |

Specifically, we fine-tune OmniVinci on the training split of R2R-CE (Krantz et al., 2020), a benchmark for Vision-and-Language Navigation in continuous environments, with speech prompts, using 8 history frames for context in line with NVILA (Liu et al., 2025a). As shown in the results in Table 9, OmniVinci surpasses many text-based models and achieves performance comparable to NVILA. We present qualitative examples in Figure 8 that illustrate how our speech-driven vision-language-action (VLA) navigation agent functions in practice. The agent is deployed in the Habitat simulator under the continuous environment setting. The demo provides three synchronized views: (1) the agent's current observation in RGB (left), (2) a top-down map indicating the goal location and the trajectory taken so far (middle), and (3) the spoken instruction together with the agent's predicted action, such as moving forward a certain distance or turning left or right by a specified angle (right).

## C.2 SPORT VIDEO UNDERSTANDING

Understanding videos of complex sports scenarios requires models to capture both visual dynamics and contextual cues. To evaluate the sports understanding capability of our proposed OmniVinci, we conduct experiments on the SPORTU-video dataset (Xia et al., 2025), a large-scale benchmark for fine-grained sports comprehension. As shown in Table 10, OmniVinci-9B delivers strong performance despite its compact scale of 9 billion parameters. These results confirm the effectiveness of our model

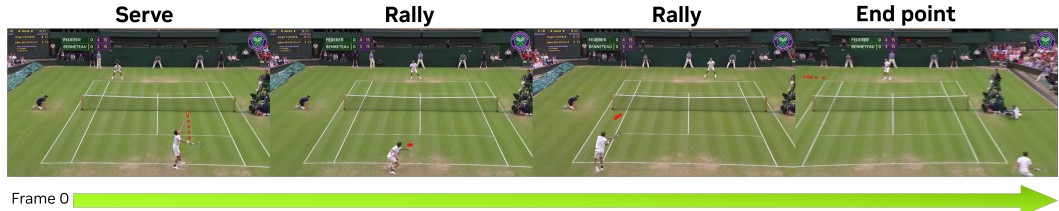

**System Prompt:** Federer – Roger Federer is the player in the white polo T-shirt and matching shorts. He has a white bandanna tied around his forehead and a white wristband on his right wrist. He is also wearing a pair of white shoes with white socks. Benneteau – Julien Benneteau is the player in the white polo T-shirt with black accents and matching shorts. He has a white wristband on his right wrist and a white pair of shoes with black markings on them. He has a black ankle brace on his right foot as well.

**Instruction:** What happened in this point? Provide a detailed caption.
**Agent:** Benneteau's first serve lands outside the service court, but his second attempt is successful. Federer wins the point with a lengthy rally where Benneteau hits the ball out of bounds with a forehand shot.

**Instruction:** How this point ended between Federer and Benneteau.
**Agent:** The point ended when Benneteau hit the ball out of bounds.

Figure 9: Example of tennis broadcast commentary generation. For better visualization, we added red circle highlights to the tennis ball.

Table 10: Overall performance of MLLMs on SPORTU-video for multiple-choice questions. The best results within each category are **bolded**. Notably, our OmniVinci model achieves highly competitive accuracy compared with both closed-source and open-source models.

| Model | (Acc., ↑) |
|---|---|
| *Close-Source Model* | |
| Claude-3.0-Haiku | 47.95 |
| Gemini 1.5 Pro | 64.93 |
| Gemini 1.5 Flash | 62.52 |
| GPT-4omini | 58.19 |
| GPT-4o | **68.79** |
| *Open-Source Model* | |
| ChatUniVi | 41.89 |
| LLaVA-NeXT | 63.72 |
| mPLUG-Owl3 | 60.80 |
| ST-LLM | 46.39 |
| Tarsier | 60.99 |
| Video-ChatGPT | 34.05 |
| VideoChat2 | 61.53 |
| Qwen2.5-Omni-7B | 60.49 |
| OmniVinci-9B (ours) | **67.30** |

design and motivate its extension to more demanding, real-world applications such as live sports broadcasting, where both accuracy and efficiency are essential.

To further assess performance in realistic broadcasting settings, we curate a tennis-specific dataset collected from 14 full matches. The dataset contains 24,078 multiple-choice questions and 20,214 open-ended questions derived from pre-clipped videos, each spanning 3–120 seconds with precisely annotated start and end points. Since sports broadcasting requires synchronizing visual actions with speech cues (*e.g.*, live commentators' narration or umpire calls) to enable professional-style commentary, tennis provides an ideal domain for multimodal evaluation.

In our tennis experiments, we evaluate tasks such as identifying the server from player characteristics, determining the point winner, and classifying the outcome type (*e.g.*, ace, forced error, unforced error). The benchmark OmniVinci processes clips at their native resolution (primarily FHD $1920 \times 1080$), using 128-frame segments per point. As shown in Table 11, OmniVinci substantially outperforms Qwen2.5-Omni in predicting point outcomes and rally length, demonstrating the advantages of high-resolution spatiotemporal modeling. Figure 9 illustrates sample videos with action explanations, along with generated open-ended commentary styled after professional broadcasters.

Table 11: Comparison of video understanding accuracy (%) for tennis broadcasting. Results are evaluated with multiple-choice questions (MCQ). Inference time is measured on an NVIDIA A100, with input clips averaging around 20 seconds in duration. AWQ indicates model quantization performed with the AWQ technique (Lin et al., 2024a).

| Model | Inference Time (Seconds ↓) | Server & Winner | Receiver & Winner | Point Ending | Shots Exchanged |
|---|---|---|---|---|---|
| Qwen2.5-Omni | 3.34 | 96.2 | 90.7 | 48.6 | 38.3 |
| **OmniVinci** | 3.29 | **100.0** | **100.0** | **85.7** | **89.3** |
| **OmniVinci w/ AWQ** | **1.85** | **100.0** | **100.0** | **85.7** | 85.1 |

Table 12: Performance comparison of different models on Covost2 speech translation tasks measured by BLEU scores. EN → X denotes translation from English to the target language, and X → EN denotes translation from the target language to English. Languages: zh = Chinese, ja = Japanese, ar = Arabic, de = German.

| Model | EN → X (Acc., ↑) | | | | | X → EN (Acc., ↑) | | | | |
|---|---|---|---|---|---|---|---|---|---|---|
| | zh | ja | ar | de | avg. | zh | ja | ar | de | avg. |
| Qwen2-audio | **45.2** | 24.8 | 20.1 | 29.9 | 30.0 | 24.4 | 18.7 | 19.5 | 35.2 | 24.5 |
| Qwen2.5-omni | 41.4 | 26.0 | 19.7 | 30.2 | 29.3 | 29.4 | 12.1 | 19.3 | 37.7 | 24.6 |
| Phi-4-mm | 38.0 | 31.9 | 9.9 | 35.3 | 28.9 | 24.9 | 33.3 | 5.5 | **37.9** | 25.7 |
| **OmniVinci** | 39.7 | **32.6** | **23.3** | **35.5** | **32.8** | **29.9** | **33.7** | **20.1** | 32.6 | **29.1** |

For efficient deployment, we adopt the LLM-AWQ implementation of Activation-aware Weight Quantization (Lin et al., 2024a), which enables 4-bit quantization while preserving accuracy. Inference is executed using the TinyChat engine on NVIDIA hardware, supporting multimodal video–audio inputs. On a single NVIDIA A100, OmniVinci achieves an average latency of under 2 seconds per pre-quantized clip, delivering a 45% boost in inference speed and making it well-suited for live broadcasting scenarios. We further validate deployment on NVIDIA L40s GPUs, demonstrating the practicality of our approach in resource-constrained environments.

## C.3 Speech Agent: Speech Translation

We benchmark OmniVinci on the CoVoST2 (Wang et al., 2020) speech translation task, measuring BLEU scores across multiple target languages in both EN →X and X→EN directions, after fine-tuning on related data, and show the results in Table 12. Our model delivers competitive translation quality across most directions, with particularly strong performance in X → EN for Japanese (23.2 BLEU) and Arabic (23.0 BLEU). This balance of accuracy across languages highlights the benefit of integrating speech translation corpora within our omni-modal training pipeline, enabling to perform both recognition and translation in a unified framework. The ability to handle multilingual speech understanding and cross-lingual transfer further broadens the applicability of our model in real-world communication, international dialogue systems, and cross-border information access.

## C.4 Medical AI

We evaluate OmniVinci's zero-shot generalization to the medical domain using 49 privacy-deidentified, radiologist-curated video clips of whole-body CT interpretations. As illustrated in Figure 10, each 2-minute recording captures a radiologist interpreting real-world clinical images with a 2D axial-plane viewer, including scrolling through slices, placing measurements and annotations, zooming, adjusting window/level, and, when relevant, comparing the same image under different window settings.

From these video–audio pairs and their transcripts, we construct 588 multiple-choice questions spanning four categories—(i) long-horizon temporal reasoning and localization, (ii) audio–visual synchronization and understanding, (iii) anti-shortcutting (resisting language priors without visual evidence), and (iv) temporal reasoning—approximately balanced across categories with three options per item. The dataset was curated with assistance from the LLama-3.1-Nemotron-Ultra-253B (Bercovich et al., 2025), leveraging both the visual content and transcripts. We report comparative performance for OmniVinci and Qwen2.5-Omni in Table 13.

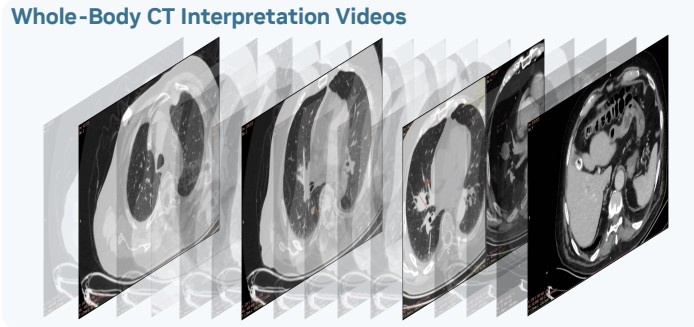

Figure 10: Sample frames and transcript trunks from one of the curated radiologist-narrated CT interpretation video. For annotation, the radiologist maintains a 2D axial view while progressively adjusting visualization (*e.g.*, window/level, zoom) and annotating across slices.

Table 13: Performance comparison between OmniVinci and Qwen2.5-Omni on omni-modal multiple-choice QA datasets across four categories. Abbreviations: LH = long-horizon temporal reasoning & localization; AVS = audio-visual synchronization & understanding; AS = anti-shortcutting (resisting language priors without video evidence); TR = temporal reasoning.

| Method | Acc. (LH) ↑ | Acc. (AVS) ↑ | Acc. (AS) ↑ | Acc. (TR) ↑ | Average ↑ |
|---|---|---|---|---|---|
| Qwen2.5-Omni | 0.83 | 0.75 | 0.91 | 0.70 | 0.79 |
| **OmniVinci** | **0.84** | **0.76** | **0.92** | **0.76** | **0.82** |

OmniVinci consistently outperformed Qwen2.5-Omni across all four categories, yielding an overall gain of about +2.0 percentage points. Its largest margin was in temporal reasoning (TR; +6.1), highlighting stronger capabilities in event sequencing, change detection, and temporal cue modeling—often the most demanding aspects of video comprehension in clinical workflows. Stable improvements were observed in long-horizon reasoning (LH) and audio-visual synchronization (AVS) (+0.7 each), reflecting better preservation of long-range context and closer alignment between narration and visual content. The anti-shortcutting (AS) category also showed a gain of +0.7, suggesting that OmniVinci is more robust against linguistic shortcuts and leans more heavily on visual evidence. Some qualitative comparisons of test samples are presented in Figure 11.

## C.5 Smart Factory Agents

### C.5.1 Semiconductor Manufacturing

Wafer maps are essential in semiconductor manufacturing for visualizing defect distributions, enabling yield monitoring, process drift detection, and preliminary root cause identification. It is a domain with a significant gap from multimodal LLM. To study whether we can leverage our omni-modal OmniVinci on this task, we fine-tune OmniVinci on wafer map data, aligning visual and textual features for robust defect analysis, as illustrated in Figure 12. On the WM-811K dataset (Wu et al., 2015), OmniVinci achieves superior performance over VILA (Lin et al., 2024b) and NVILA (Lin et al., 2025a; Liu et al., 2025a) (which has been trained for wafer defect classification), and our model demonstrates further improvements, as summarized in Table 14. Beyond classification, this framework can be extended to support interactive querying and automated reasoning for Root Cause Analysis, systematically linking defect clusters to process tools, wafer locations, or temporal drifts.

| Long-horizon temporal reasoning & localization | Audio-visual synchronization & understanding |
|---|---|
| You will be asked multi-choice questions. Your replies must contain only a single letter (either A, B, C, D). If each subtle intensity change in the lung fields represents a 5% adjustment in diagnostic confidence for pneumonia, how many such changes occur from 0 to 120 seconds, requiring tracking across the entire video duration?

A.  4 adjustments (20% to
B.  10 adjustments (50% total)
C.  6 adjustments (30% total)
D.  8 adjustments (40% total)'}

Ground truth: **B**
Qwen2.5-Omni: **C**
Ours: **B** | You will be asked multi-choice questions. Your replies must contain only a single letter (either A, B, C, D). What structure is highlighted by the green circular marker added near the lung area at 20-30 s?

A.  Spine
B.  Trachea
C.  Bronchus
D.  Lung nodule


Ground truth: **D**
Qwen2.5-Omni: **C**
Ours: **D** |
| **Anti-shortcutting** | **Temporal reasoning** |
| You will be asked multi-choice questions. Your replies must contain only a single letter (either A, B, C, D). How many bone lesions were identified in the thorax that would support a diagnosis of metastasis?

A. No lesions
B. Multiple lesions (>3)
C. One lesion
D. Two lesions

Ground truth: **C**
Qwen2.5-Omni: **A**
Ours: **C** | You will be asked multi-choice questions. Your replies must contain only a single letter (either A, B, C, D). How do the lung textures in the CT scan change over time, based on the visual cues?

A. They transition to uniform density
B. They become more homogeneous
C. They show increasing bright white areas
D. They display consistent heterogeneous patterns

Ground truth: **D**
Qwen2.5-Omni: **C**
Ours: **D** |

Figure 11: Qualitative comparison between OmniVinci and Qwen2.5-Omni on an omni-modal medical QA task based on radiologist-narrated CT interpretation videos. We organize the evaluation into four categories of questions: long-horizon temporal reasoning and localization, audio-visual synchronization and understanding, anti-shortcutting, and temporal reasoning.

**User**: This is a image of a wafer map, the yellow pattern in the circle refers to the defect pattern. There are 8 possible types of defect of wafer map (1) loc. (2) edge-loc. (3) center. (4) edge-ring. (5) scratch. (6) near-full. (7) donut. (8) random. What type of anomaly does the provided image present?

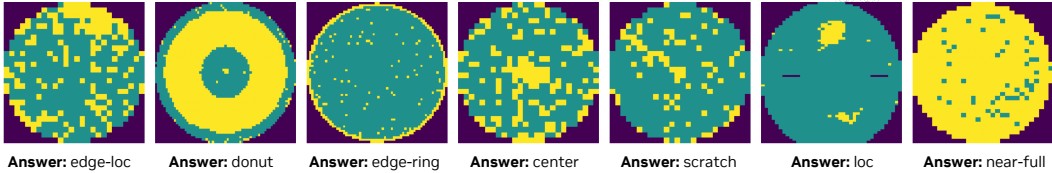

**Answer:** edge-loc   **Answer:** donut   **Answer:** edge-ring   **Answer:** center   **Answer:** scratch   **Answer:** loc   **Answer:** near-full

Figure 12: Illustration of wafer robust defect analysis task for smart factory agent.

### C.5.2 FACTORY AND INDUSTRIAL TIME SERIES UNDERSTANDING

We apply OmniVinci to Statistical Process Control (SPC) chart recognition, a representative task in industrial quality monitoring and root cause analysis. Our model recognizes a wide range of fault categories, including out-of-control points such as spikes or drops, persistent runs and monotonic trends such as level shifts up or down, cyclic oscillations, mixture or random fluctuations, as well as missing values or short outages, as illustrated in Figure 13. On a held-out test set, our model achieves 87% accuracy, showing that by transforming time-series signals into visual representations, we can effectively leverage large-scale vision-language pretraining for sensor monitoring and industrial diagnostics. This demonstrates the feasibility of deploying our framework in real manufacturing pipelines, where timely detection of process abnormalities is crucial for preventing defects and reducing downtime.

We assess our framework on time series classification tasks using datasets from the UCR archive (Dau et al., 2018), where time series are transformed into line plots to exploit large-scale vision–language pretraining. Our first comparison is against VLM-TSC (Prithyani et al., 2025), a LLaVA-based VLM that adopts a similar conversion strategy. As shown in Table 15, our approach achieves superior performance on the PenDigits and ItalyPowerDemand datasets.

Table 14: Comparison of VILA, NVILA, and OmniVinci on wafer defect classification.

| | **VILA** (Lin et al., 2024b) | **NVILA** (Liu et al., 2025a) | **OmniVinci** (ours) |
|---|---|---|---|
| Parameters | 40B | 8B | 9B |
| Resolution | 336×336 | 448×448 | 448×448 |
| Model size | 75 GB | 16 GB | 18 GB |
| Accuracy | 90.8% | 97.6% | **98.1%** |

**User**: What class do these images belong to? The possible classes are: cluster, constant, cycling, missing, period_trending, periodic_patterns, shift, trending, uneven.

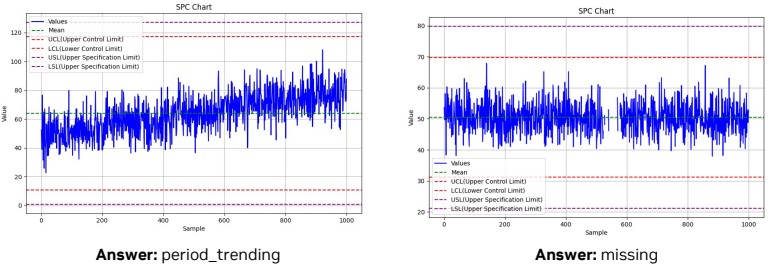

**Answer:** period_trending          **Answer:** missing

Figure 13: Illustration of SPC chart recognition for industrial fault detection.

# D    METHOD DETAILS

## D.1    OMNI-MODAL INPUT EMBEDDING

**Image.** Similar to NVILA (Liu et al., 2025a), we start with pretrained the SigLip (Zhai et al., 2023) [1] vision encoder and augment it with $2 \times 2$ "Spatial Scale-Then-Compress" Dynamic S2 (Liu et al., 2025a; Shi et al., 2024) to accomodate for multi-scale and high resolution images. Given an input image of varying dimensions, the overall encoding module adapts the largest scale to the nearest tile-aligned size divisible by 448 and maintains the aspect ratio. Feature maps from all scales are aligned to this largest scale and concatenated, processed by a 2-layer MLP for projection into a latent space shared by embeddings of different modalities.

**Audio.** We adopt a single audio encoding pipeline for both speech and non-speech audio. Raw audio waveforms are sampled at 16 kHz and converted into audio frames using the Short-Time Fourier Transform (STFT). These frames are then processed by the Audio Flamingo 3 (Goel et al., 2025) audio encoder to extract acoustic features in both speech and natural sound. The encoder consists of convolutional layers followed by transformers, enabling it to capture both local and global audio patterns. The extracted features are subsequently projected into the modality-shared latent space using a 2-layer MLP.

**Video.** Videos contain two modalities introduced above, namely vision and audio. For the vision stream, the video frames are temporally sampled uniformly to reduce redundancy and computational load. Each frame is processed individually through the above-mentioned image input pipeline, and the resulting features are aggregated temporally. We then utlize temporal pooling on the feature sequence to further compress visual information. For audio stream, we extract features with the same Audio modality pipeline mentioned above. Meanwhile, we extract the timestamps for each visual and audio embeddings to act as temporal guidance on interleaved token arrangement as explained later.

**Prompt.** For text prompts, we employ a standard text encoder (Qwen, 2024), which first tokenizes the input into discrete tokens and then maps them into a continuous semantic embedding space via an embedding layer. This space is shared with embeddings from other modalities. For speech prompts, we use the previously described audio encoder to generate the corresponding continuous semantic embeddings. Finally, the resulting prompt embeddings are concatenated with the visual and audio embeddings introduced earlier.

---

[1]Model version "`paligemma-siglip-so400m-patch14-448`"

Table 15: Performance comparison of test accuracy (%) on selected UCR datasets (Dau et al., 2018).

| Dataset Info | | | | | | Acc. ↑ | |
|---|---|---|---|---|---|---|---|
| Dataset | Type | Length | Train | Test | Class | VLM-TSC (Prithyani et al., 2025) | Ours |
| PenDigits | MOTION | 8 | 7494 | 3498 | 10 | 85.08 | **96.88** |
| ItalyPowerDemand | SENSOR | 24 | 67 | 1029 | 2 | 95.00 | **95.82** |

## D.2 MORE DISCUSSION ON CONSTRAINED ROTARY TIME EMBEDDING (CRTE)

The base frequency in CRTE, $\omega_i$ is designed to have a geometric progression of frequencies. For small values of $i$ (*e.g.*, the first pairs of dimensions), the denominator is smaller, resulting in higher frequencies ($\omega_i$ is large). These dimensions undergo rapid rotation with respect to time. Consequently, they are highly sensitive to fine-grained temporal differences and are effective at distinguishing between timestamps that are close to one another. For large values of $i$ (*e.g.*, the last pairs of dimensions), the term $\theta^{i/d}$ becomes significantly larger, resulting in lower frequencies ($\omega_i$ is small). These dimensions rotate slowly, making them suitable for encoding coarse, long-range temporal relationships. They provide a stable signal for large time intervals without the issue of aliasing or "wrapping around" that would occur with high-frequency signals. By partitioning the embedding space into a spectrum of frequencies, the model can concurrently attend to both local and global temporal contexts. This multi-scale approach provides a robust and comprehensive representation of absolute time.

## D.3 MODALITY-SPECIFIC TRAINING

### D.3.1 VISION TRAINING

The modality-specific vision training aims to train the model with visual understanding ability. We follow NVILA (Liu et al., 2025a) training recipe including five stages:

**Stage 1 | Vision Projector Alignment.** This stage learns to project visual information through a visual projector. This stage ensures that the visual embeddings are compatible with the language model's token embeddings, which is essential for smooth downstream integration. The model is trained on image-text pairs with simple captioning-style supervision, setting a baseline understanding of visual semantics. Only the vision projector is tuned during this process.

**Stage 2 | Vision Encoder Alignment.** With the projector aligned, the model now focuses on enhancing the vision encoder's capacity to process diverse visual content. In this stage we train only the vision encoder and visual projector.

**Stage 3 | Vision Pre-Training.** During this core stage, the model is trained on large-scale multimodal data to learn how to interpret and generate image descriptions. The vision encoder is kept frozen, while the vision projector and the LLM are fine-tuned.

**Stage 4 | Image Instruction Tuning.** In this stage the model is fine-tuned with vision instruction-following capabilities. It is trained to answer multimodal questions, generate captions, reason over scenes, interpret documents, and more. Training data covers a broad range of multimodal capabilities. It includes high-quality instructional examples to align the model with human preferences, datasets for generating rich image captions, and tasks that develop logical and visual reasoning skills. The model is also trained to interpret documents and embedded text, answer general and knowledge-based visual questions, and handle diagrams, visual dialogues, and multimodal instructions. In this stage, all model parameters are fine-tuned.

**Stage 5 | Video Instruction Tuning.** In the final vision alignment stage, the model is adapted to video understanding. The goal here is to enable temporal reasoning and visual understanding over sequences of frames. This includes tasks such as activity recognition, multi-frame object tracking, and answering time-sensitive questions. The whole model is fine-tuned.

Through this vision alignment process, we obtain the "vision preliminary checkpoint" with well-trained vision encoder, projector, and language model.

### D.3.2 AUDIO TRAINING

Starting from the language model in the above vision preliminary checkpoint we next train the audio understanding ability of our model, which involves (i) audio projector and encoder alignment step followed by (ii) audio instruction tuning.

**Stage 1 | Audio Projector & Encoder Alignment.** This phase focuses on aligning audio encoder and its associated compression layer. We keep the parameters of the language model and vision side fixed. Training consumes 50K audio-language pairs curated from public datasets spanning across audio-based (music, non-speech sound, and speech) question answering, speech-to-text captioning, and automatic speech recognition. By training on this heterogeneous dataset, we encourage the audio projection module to learn a unified representation that aligns well with the language model's semantic space.

**Stage 2 | Audio Instruction Tuning.** During the second stage of training, the audio encoder, audio projection module, and language model are fine-tuned in a unified, end-to-end manner. This joint optimization allows the system to develop a comprehensive and deeply integrated understanding of audio. This stage consumes a comprehensive audio-SFT dataset overseeing 9.6 million samples, including but not limited to audio-based question answering (AudioEntailmentQA (Deshmukh et al., 2025), Clotho-AQA (Lipping et al., 2022), DCASE-2025-train (Yang et al., 2025b), etc.), audio captioning (AudioCaps (Kim et al., 2019), Clotho-v2 (Drossos et al., 2020), Miradata (Ju et al., 2024)-recaptioned, etc.), speech emotion recognition (CREMA-D (Cao et al., 2014), IEMOCAP (Busso et al., 2008), MELD (Poria et al., 2018), etc.), automatic speech recognition (CV-ASR (Ardila et al., 2020), Europarl-ASR (Koehn, 2005), LibriSpeech-ASR (Panayotov et al., 2015), etc.), and speech translation (MuST-C (Di Gangi et al., 2019), Emilia (He et al., 2024), etc.). This allows the model to learn both low-level acoustic features and high-level semantic representations, enabling robust generalization across multiple audio understanding tasks and versatile capabilities in interpreting complex auditory inputs. At this point, we find that the model's ability to perform visual understanding tasks is worse, which motivates us to pursue the subsequent omni-modal joint training.

### D.4 OMNI-MODAL JOINT TRAINING DETAILS

We adopt a cosine learning rate schedule, preceded by a linear warm-up phase over the first 3% of the training data. The base learning rate is set to $2 \times 10^{-5}$. During training, the vision and audio encoders are kept frozen. The total token count is approximately 200 billion.

### D.5 EXTRA DETAILS OF TRAINING DATA

This section describes the comprehensive multi-modality training data used for developing the proposed omni-modal LLM, which are designed to handle diverse types of audio, visual, and textual information. Our training corpus encompasses a wide range of modalities including speech recognition, audio question answering, audio captioning, audio classification, video question answering, and image understanding tasks. The dataset is carefully curated to provide robust coverage across multiple domains, enabling the model to develop strong cross-modal understanding and reasoning capabilities.

There are 3.6 million omni-modal conversations, 8 million image-text conversations, 2.7 million video-text conversations, 5.3M speech-text conversations, and 4.3 million speech-text conversations. Omni-modal data contributes 15%, consisting of omni question answering (12%) and omni captioning (3%). Image data constitutes the largest share at 36%, with notable subcategories including general image tasks (19%), knowledge-based tasks (8%), and document processing (7%). Sound (non-speech) data accounts for 21%, predominantly driven by audio question answering (20%). Speech data represents 17% of the total, primarily comprising automatic speech recognition (10%), audio question answering (4%), and speech translation (2%). Video data forms the remaining 11%, entirely attributed to video question answering. The training data consists of approximately 24 million samples distributed across three main categories: Speech, Sound, and Image/Video. The Speech category includes datasets for automatic speech recognition (ASR), speech translation, and emotion classification, featuring well-established corpora such as AMI (Carletta, 2007), Common Voice (Ardila et al., 2020), and LibriSpeech (Panayotov et al., 2015). The Sound category encompasses audio question answering datasets like MMAUQA (Goel et al., 2025) and CompA-R-AQA (Ghosh et al., 2024),

Table 16: MMAU audio benchmark.

| Model | Music | | Sound | | Speech | | Avg | |
|---|---|---|---|---|---|---|---|---|
| | Test | Test-mini | Test | Test-mini | Test | Test-mini | Test | Test-mini |
| Gemini 2.5 Pro | 68.26 | 64.77 | 70.63 | 75.08 | 72.67 | 71.47 | 71.60 | 69.36 |
| Gemini 2.5 Flash | 76.58 | 69.40 | 65.57 | 69.50 | 71.80 | 68.27 | 69.57 | 67.39 |
| Kimi-Audio | 62.16 | 65.93 | 66.77 | 70.70 | 56.57 | 68.20 | 64.40 | 68.20 |
| Phi-4-multimodal | 61.97 | 64.37 | 62.67 | 65.47 | 63.80 | 67.27 | 62.81 | 65.70 |
| Audio Flamingo 2 | 44.74 | 70.20 | 68.13 | 70.96 | 44.87 | 62.40 | 61.06 | 62.40 |
| GPT-4o Audio | 49.93 | 56.29 | 63.20 | 64.56 | 69.33 | 66.67 | 60.82 | 62.50 |
| Qwen2-Audio-Instruct | 55.26 | 55.67 | 56.29 | 61.17 | 59.60 | 55.37 | 57.40 | 59.60 |
| Gemma 3n 4B | 61.26 | 53.20 | 56.89 | 50.27 | 58.00 | 62.13 | 58.00 | 55.20 |
| Qwen2.5-Omni | 67.33 | 65.90 | 76.77 | 78.10 | 68.90 | 70.60 | 71.00 | 71.50 |
| **OmniVinci** | 73.07 | 73.65 | 73.57 | 78.68 | 68.17 | 66.97 | **71.60** | **73.10** |

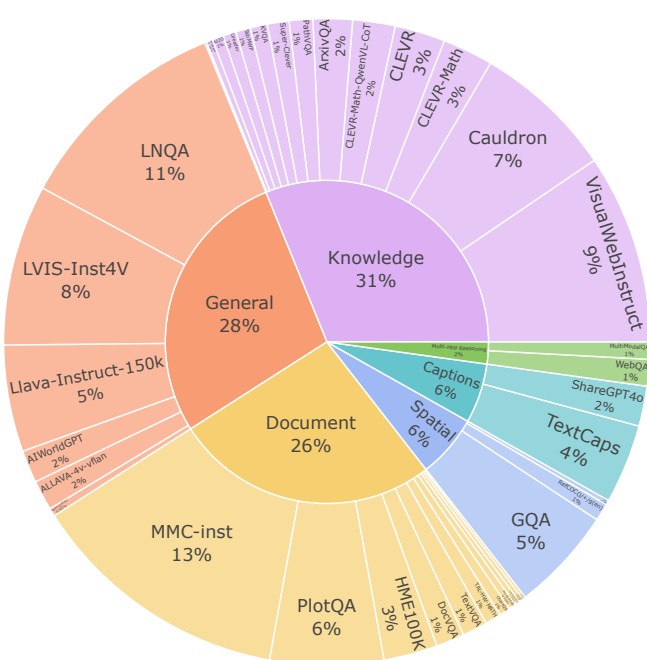

Figure 14: Data distribution of our synthetic speech-prompted multimodal conversation.

audio captioning datasets such as Clotho-v2 (Drossos et al., 2020), and various audio classification datasets including UrbanSound8K (Mesaros et al., 2018) and FSD50k (Fonseca et al., 2021). The Image/Video category includes datasets for visual question answering, document understanding, and general image comprehension tasks.

# E    MORE EXPERIMENTS AND DISCUSSION

## E.1    AUDIO ENCODING

**Audio Encoder Backbone.** To investigate the choice of audio representations for the omni-modal model, we evaluate two state-of-the-art audio encoders: Qwen2-Audio (Chu et al., 2023) used by Qwen2.5-Omni (Xu et al., 2025), and the AF-Whisper backbone (Goel et al., 2025) from Audio Flamingo 3 (Goel et al., 2025). This comparative analysis enables us to identify the backbone that provides the most effective encoding for downstream multimodal tasks. Specifically, we ablate these key components by aligning them with the LLM backbone model we used in audio-only training. We use 10% of the audio/speech training data to fairly evaluate the effectiveness of the two encoders

under the same data budget. As shown in Table 17, AF-Whisper consistently outperforms the Qwen-2 Audio encoder backbone on audio and speech understanding tasks. Therefore, our final model architecture adopts the AF-Whisper backbone to extract informative audio features.

Table 17: Ablation study on different Audio Encoder backbones.

| Audio Encoder | LS-clean | LS-other | MMAU-mini | MMAU |
|---|---|---|---|---|
| Qwen2-Audio | 5.5 | 7.1 | 61.5 | 59.0 |
| AF-Whisper – **chosen** | **2.1** | **5.2** | **70.5** | **63.3** |

**Audio Token Compression.** For the AF-Whisper encoder, similar to Whisper-large-v3 (Radford et al., 2023), the process begins by resampling the audio to a 16 kHz sampling rate, followed by transforming the raw waveform into a 128-channel mel-spectrogram using a 25 ms analysis window and a 10 ms hop interval (*i.e.*, a hop length of 160). This yields 3,000 audio frames for a 30-second audio, which are then processed through convolutional layers and a transformer model to extract audio features, resulting in 750 sequential audio feature vectors. Therefore, each second of audio is roughly represented by 25 tokens. While this may not seem like a lot for a 30-second audio, encoding one hour of audio would require about 90,000 tokens, which could overwhelm the context length of multimodal models.

We next explore several audio information compression strategies to improve efficiency in representing audio information. In our ablation study, we fine-tune the preliminary checkpoint before large-scale training on a 2.6M audio-only dataset, referring to this configuration as the *Baseline*. We then evaluate two audio feature compression methods: (i) Applying 1-D convolution with kernel size 3 and stride 2 before audio projector, or (ii) Applying average or max pooling with kernel size 2 before audio projector. We assess performance on audio understanding benchmarks, including Librispeech, Gigaspeech, VoxPopuli, and Long Audio Bench (Goel et al., 2025) and present results in Table 18. We also report the embedding per minute of input audio and the average end-to-end latency of the LLM forward pass on Long Audio Bench for each variant in the table.

Table 18: Downsampling method comparison for audio token compression in OmniVinci. For Librispeech, Gigaspeech, and VoxPopuli we report WER (lower is better). For Long Audio Bench we report accuracy (higher is better) and latency (lower is better). Gains are computed relative to the baseline (All audio tokens).

| Model | Emb./min ($\downarrow$) | Librispeech-cl. WER ($\downarrow$) | Librispeech-oth. WER ($\downarrow$) | Gigaspeech WER ($\downarrow$) | VoxPopuli-ASR WER ($\downarrow$) | Long Audio Acc. ($\uparrow$) | Long Audio Lat. ($\downarrow$) |
|---|---|---|---|---|---|---|---|
| Baseline - All audio tokens | 750 | 1.91 | 4.49 | 10.77 | 5.89 | 41.28 | 1.78 |
| **Audio Compression** | - | - | - | - | - | - | - |
| Conv1D stride 2 | 375 | $2.10_{-0.19}$ | $5.22_{-0.73}$ | $11.01_{-0.24}$ | $6.25_{-0.36}$ | $41.79_{+0.51}$ | $1.45_{+0.33}$ |
| Avg. pooling | 375 | $\underline{1.96}_{-0.05}$ | $\underline{4.75}_{-0.26}$ | $\underline{10.85}_{-0.08}$ | $\underline{6.24}_{-0.35}$ | $\underline{42.16}_{+0.88}$ | $1.41_{+0.37}$ |
| Max pooling – **chosen** | 375 | $\mathbf{1.93}_{-0.02}$ | $\underline{4.99}_{-0.50}$ | $\mathbf{10.78}_{-0.01}$ | $\mathbf{6.17}_{-0.28}$ | $\mathbf{43.15}_{+1.87}$ | $\mathbf{1.40}_{+0.38}$ |

We observe several advantages via compression. Halving audio tokens leads to significantly shorter latency, from 1.78 sec/sample to 1.40 sec/sample (+17.7% improvement). For the long audio understanding task, applying audio token downsampling improves the accuracy by 2% as it compresses information into a more condense representative embeddings, alleviates the burden on LLMs when handling large volumes of audio embeddings. For short-form benchmarks, we study varying downsampling options, where we observe max pooling maintains performance across benchmarks without minimal accuracy degradations.

## E.2  MODEL QUANTIZATION AND EFFICIENT DEPLOYMENT

Although OmniVinci demonstrates strong omni-modal performance, real-world deployment quickly encounters multiple constraints. Large models or long video sequences often exceed device memory capacity, while interactive applications demand extremely low latency. To meet these challenges, we compress the model via quantization and optimize the system for speedup. A detailed analysis of the inference pipeline reveals distinct bottlenecks: the vision and audio towers are dominated by dense matrix multiplications, processing large batches of tokens in parallel and thus primarily computation-bound; in contrast, the LLM decoding stage—where the model consumes and generates one token at a time— is memory-bandwidth limited and becomes the key latency bottleneck in

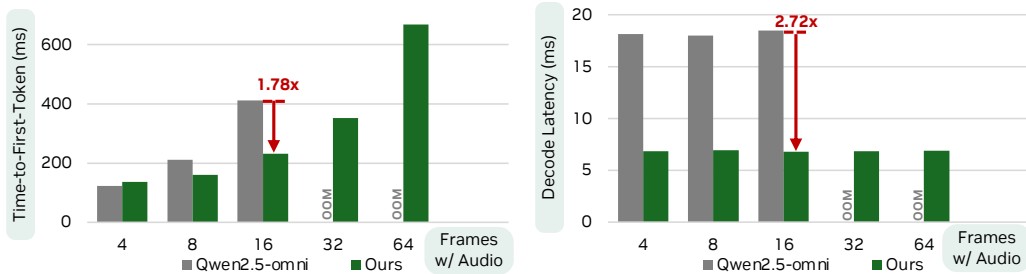

Figure 15: Latency comparison between Qwen2.5-Omni and our OmniVinci model on a GeForce RTX 4090 GPU. Our model achieves $1.7\times$ faster time-to-first-token latency and $2.72\times$ faster decoding latency.

long-context scenarios. To address this, we adopt a component-aware quantization strategy. For the vision and audio towers, we apply W8A8 quantization, reducing arithmetic cost while preserving representational quality. For the LLM, we employ W4A16 quantization, compressing weights into 4 bits while retaining 16-bit computation, which accelerates bandwidth-limited decoding. Finally, to recover accuracy, we integrate Activation-Aware Weight Quantization (AWQ) (Lin et al., 2024a) and SmoothQuant (Xiao et al., 2023).

We measure the time-to-first-token latency and decoding latency on a single GeForce RTX 4090 GPU using video clips ranging from 2 to 32 seconds (at 2 frames per second), and compare the performance against Qwen2.5-Omni in Figure 15. Overall, these quantization methods allow a 8B model to handle videos of up to 64 frames on a 24GB RTX 4090 GPU, while achieving $1.7\times$ lower time-to-first-token latency and $2.72\times$ faster decoding latency. For a 16-frame video with audio stream, our model needs only around 160ms to produce the first token.

### E.3 OMNIVINCI WITH ASR TEST-TIME SCALING METHODS

To push the limit of transcription accuracy, we investigate our model's ability to leverage pre-trained ASR models in downstream speech understanding tasks. In a cascaded post-ASR processing setup (Yang et al., 2023) as shown in Figure 16 (a), speech inputs are first transcribed by the model's ASR module and then processed by LLM based generative ASR error correction. We use a popular 800M streaming variant of Whisper-v3-Turbo from SimulStreaming as the cascaded ASR module.

The results are also shown in Table 19. The cascaded pipeline yields additional improvements on ASR tasks, making it particularly beneficial in offline transcription scenarios. We use Phi-4-mm-instruct 's 5-shot (Abouelenin et al., 2025) speech modeling setup as one test-time baseline. For Qwen2.5-Omni experiment, we follow the official inference script[2] for the evaluation reported in the fourth row of Table 19, with the original results shown in the third row. From the extended test-time scaling results, OmniVinci-cascaded improves average WER from 6.3 to 5.7. The OmniVinci-RAG setup yields a further improvement, reducing average WER from 6.3 to 5.0 with the same model size of ASR parallel cascading by using ASR text as index for OmniVinci on mutlimodal ASR correction (Lin et al., 2025b). We introduce the retriever training details of this setup in the following section.

**OmniVinci with ASR based Retriever-Augmented Training.**

As shown in Figure 16 (b), given a primary acoustic input, $\mathcal{A}$, our objective is to generate a final, high-fidelity textual output $\mathcal{T}_{\text{final}}$ (either a transcription for ASR or a translation for ST). The model has access to two streams of textual information:

1. **Internal Hypothesis** ($\mathcal{T}_{\text{internal}}$): A first-pass generation produced by the omni-modal model itself, conditioned solely on the acoustic input $\mathcal{A}$. This represents the model's direct, audio-grounded interpretation.

---

[2]We follow the official ASR cookbook in `https://github.com/QwenLM/Qwen2.5-Omni/blob/main/cookbooks/universal_audio_understanding.ipynb` and a related discussion in `https://github.com/QwenLM/Qwen2.5-Omni/issues/79` on the Omni settings used in our evaluation.

Table 19: Speech Recognition WER (%) comparison of different models on speech recognition datasets.

| Model | WER (↓) | | | | | |
|---|---|---|---|---|---|---|
| | LS$_{\text{clean}}$ | LS$_{\text{other}}$ | AMI | Tedlium | Voxpopuli | Avg. |
| Phi-4-MM | 1.7 | 3.8 | **11.5** | **2.9** | 5.9 | 5.2 |
| Phi-4-MM-in-context (5-shots) | 1.6 | 3.6 | **11.5** | 3.0 | 6.1 | 5.2 |
| Qwen2.5-omni: reported (Xu et al., 2025) | 1.8 | 3.4 | - | - | 5.8 | - |
| Qwen2.5-omni: reproduced | 2.1 | 3.8 | 17.8 | 5.2 | 6.1 | 7.0 |
| **OmniVinci** | 1.7 | 3.7 | 16.1 | 3.4 | 6.8 | 6.3 |
| **OmniVinci**-cascaded | 1.6 | **3.0** | 14.1 | 3.3 | 6.5 | 5.7 |
| **OmniVinci**-RAG | **1.5** | **3.0** | 11.6 | 3.0 | **5.7** | **5.0** |

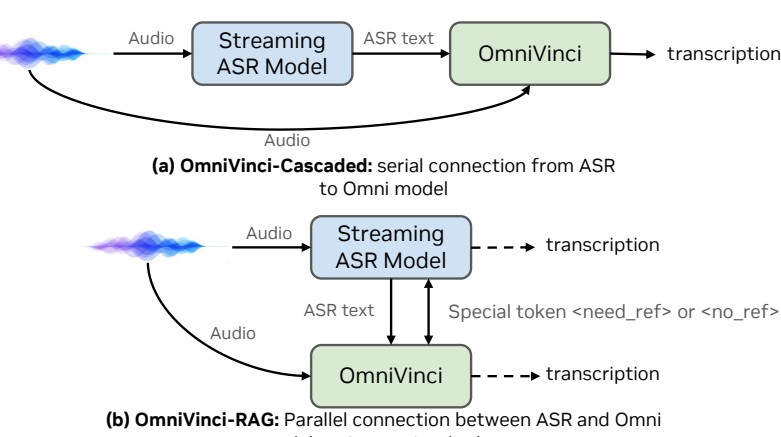

(a) **OmniVinci-Cascaded:** serial connection from ASR to Omni model

(b) **OmniVinci-RAG:** Parallel connection between ASR and Omni models using retrieval token

Figure 16: We illustrate two test-time scaling methods using an extra ASR model: (a) OmniVinci-Cascaded, using ASR history as an additional input to the Omni model with the audio inputs, and (b) OmniVinci-RAG, using the retrieval token for prediction. The related results are reported in Table 19.

2. **External References ($\mathcal{H}$):** A set of candidate transcriptions, $\mathcal{H} = \{h_1, h_2, \ldots, h_N\}$, generated by one or more external systems. This set represents external, text-only evidence that may contain valuable corrections or introduce noise.

The task is formulated as a conditional generation problem that jointly models the final output and a decision variable, $d$. The model learns to generate a control token indicating its strategy, followed by the refined text. This decision hinges on the model's ability to align $\mathcal{A}$, $\mathcal{T}_{\text{internal}}$, and $\mathcal{H}$ to determine the most reliable path to the ground truth, $\mathcal{T}_{\text{gt}}$.

### E.3.1 INSTRUCTIONAL FORMATTING FOR CROSS-MODAL DECISION MAKING

To facilitate this decision-making process, we structure the input as a comprehensive instruction that forces the model to weigh different sources of evidence. The model is presented with all modalities and explicitly prompted to declare its generation strategy.

```
Task: Perform reference-augmented correction for a given
speech input.
Objective: Evaluate the quality of an internally generated
hypothesis against external candidates. First, select a
generation strategy by producing a control token. Then,
generate the final, corrected text.
- If the internal hypothesis is deemed superior and
well-aligned with the audio, select <accept_internal>.
- If the external candidates provide necessary corrections,
select <integrate_reference>.
--
```

```
Acoustic Evidence:  [AUDIO]
External Candidate Transcriptions:
1.  {h_1}
2.  {h_2}
3.  {h_3}
4.  {h_4}
5.  {h_5}
Internal Hypothesis:
{T_internal}
--

Output:
```

The model is then trained to generate the complete target string, beginning with either `<accept_internal>` or `<integrate_reference>`, followed by the corrected and finalized text. We expand the model's vocabulary with these two special tokens to serve as explicit control signals.

### E.3.2 SUPERVISION FOR DECISION-AWARE FINE-TUNING

Supervision for this decision-aware fine-tuning is derived by comparing the internal hypothesis ($\mathcal{T}_{\text{internal}}$) against the ground truth ($\mathcal{T}_{\text{gt}}$) and the external references ($\mathcal{H}$). The decision label is determined as follows:

- **`<accept_internal>`:** This label is assigned when the word error rate (WER) of $\mathcal{T}_{\text{internal}}$ is below a predefined threshold or when the external references in $\mathcal{H}$ offer no improvement or introduce hallucinations. This teaches the model to trust its own cross-modal alignment between audio and text when its confidence is high.
- **`<integrate_reference>`:** This label is assigned when $\mathcal{T}_{\text{internal}}$ contains correctable errors and at least one hypothesis in $\mathcal{H}$ provides information that reduces the WER relative to $\mathcal{T}_{\text{gt}}$. This trains the model to identify valuable external information and integrate it, effectively re-aligning its understanding based on supplementary textual evidence.

The final training target is the concatenation of the assigned decision token and the ground-truth transcript $\mathcal{T}_{\text{gt}}$.

### E.3.3 INFERENCE-TIME CONTROL FLOW

At inference, the omni-modal model processes the multi-source input containing the audio, its internal hypothesis, and the external references. The first token generated by the model dictates the subsequent control flow:

- If the model generates `<accept_internal>`, it signals high confidence in its own audio-to-text mapping. For the final output, we can simply use its pre-computed internal hypothesis, $\mathcal{T}_{\text{internal}}$, or allow the model to regenerate it.
- If the model generates `<integrate_reference>`, it indicates that the external textual evidence is necessary for achieving a better output. The full sequence generated by the model following this token is taken as the final, corrected transcript.

This mechanism provides an interpretable and controllable framework for test-time adaptation, allowing the model to dynamically adjust its reliance on external knowledge based on the specific challenges of each input. This is critical for robust performance in both ASR, where the focus is on transcription fidelity, and ST, where a correct semantic understanding grounded in both audio and reference text is paramount for accurate translation. Table 19 presents the performance of this method, denoted as OmniVinci-RAG. It substantially improves the model's results across all speech recognition benchmarks.

### E.4 SPEECH OUTPUT

Rather than training a speech generation model from the ground up, we leverage state-of-the-art pre-trained text-to-speech (TTS) systems to produce speech in relevant scenarios, and adapt our

approach using a speech codec when needed. Our evaluation focuses on English omni-modal-in and voice-out, using two complementary metrics: mean opinion score (MOS; higher indicates greater naturalness) and TTS word error rate (WER; lower indicates higher intelligibility), the latter measured through an external ASR system. As reported in Table 20, existing off-the-shelf models already yield high-quality, neutral speech suitable for assistant-style applications. Among the back ends tested, OmniVinci-Magpie achieves the best overall balance (MOS **4.63**, WER **2.7**%), followed closely by gpt-4o-mini-tts (MOS 4.59, WER 3.1%) and Qwen-omni (MOS 4.53, WER 3.2%). OmniVinci-StableCodec delivers a competitive WER (2.9%) but with slightly reduced naturalness (MOS 4.12), highlighting that intelligibility and perceived naturalness are not always aligned. In contrast, Bark underperforms on both measures (MOS 3.32, WER 8.2%), consistent with its more stochastic synthesis approach.

**Setup.** We evaluate prompt following on VoiceBench style/control splits and conversational control tasks. We compare three prompting strategies over interleaved audio–vision contexts: (i) *Transcript prompting* (ASR→text): $[\text{aud, vis}]^{\times 3} + \text{text-prompt}$, (ii) *Native audio prompting* (encoder features): $[\text{aud, vis}]^{\times 3} + \text{aud-prompt}$, (iii) *TTS-injected prompting* (render text to speech, then encode): $[\text{aud, vis}]^{\times 3} + \text{TTS(text-prompt)}$. We also ablate prompt position: *prefix* $[\text{aud-prompt}] + [\text{aud, vis}]^{\times 3}$, *mid* $[\text{aud, vis}], [\text{aud-prompt}], [\text{aud, vis}]^{\times 2}$, and *suffix* $[\text{aud, vis}]^{\times 3}, [\text{aud-prompt}]$.

**Metrics.** We report (a) *Prompt Adherence Rate* (PAR; judged by paired preference and rubric scoring), (b) *slot accuracy* for constrained commands (names, numerals, entities), and (c) latency/efficiency (no additional ASR pass). For speech rendering quality, MOS/WER results are summarized in Table 20.

> **Key Insight 4.** (1) *Native audio prompting* is the most robust to accents, background noise, and overlapped speech; it preserves prosodic cues (rate, emphasis) that pure transcripts discard, leading to higher PAR and slot accuracy in noisy and accented conditions. (2) *Transcript prompting* is competitive on clean speech but degrades when ASR struggles on named entities or code-switched fragments. (3) *TTS-injected prompting* reduces acoustic mismatch in far-field scenarios and is effective when a consistent house voice is desired, but it transfers less speaker/style information than using the raw prompt audio. (4) Prompt *suffix* placement—immediately before the model's response—consistently outperforms prefix and mid insertion, likely due to reduced long-range interference in the attention context.

Encoding the *audio* prompt directly (no external ASR) yields the best prompt following under realistic noise/accents while lowering latency and memory by avoiding an extra ASR pass. Suffix-position audio prompts provide the strongest control.

Table 20: English naturalness MOS (higher is better) and TTS word error rate (WER; lower is better). Best per column in **bold**.

| Setup | Regime | MOS ↑ | WER (%) ↓ |
|---|---|---|---|
| Qwen-Omni | auto-regressive | 4.53 | 3.2 |
| GPT-4o-mini | – | 4.59 | 3.1 |
| OmniVinci-CozyVoice | agentic cascaded | 4.54 | 3.0 |
| OmniVinci-Bark | agentic cascaded | 3.32 | 8.2 |
| OmniVinci-StableCodec | auto-regressive | 4.12 | 2.9 |
| OmniVinci-Magpie (**chosen**) | agentic cascaded | **4.63** | **2.7** |

Beyond raw scores, we observe consistent performance across synthesis regimes. Agentic cascaded setups that decouple text planning from acoustic rendering tend to produce strong MOS and low WER in our pipeline, while auto-regressive models are competitive but show greater variance. Importantly, swapping the TTS back end does not alter OmniVinci 's language understanding or response planning; it only affects the surface realization of speech, simplifying deployment-time customization (*e.g.*, voice, rate).

For interactive agents, streaming synthesis and low perceived latency are crucial. Our chosen back ends support incremental generation, enabling prompt first-audio while the remainder of the utterance is synthesized. In production, we prioritize (i) stability on numerals, abbreviations, and named

entities, (ii) speaker consistency across turns, and (iii) graceful handling of punctuation and prosody cues from text.

## E.5 ANALYSIS: WHY DO WE NEED OMNI-MODAL LLMS?

Humans perceive the world through vision and audio simultaneously and rely on both modalities to perform many tasks. While certain academic benchmarks for visual or audio understanding may require only a single modality, we believe that developing omni-modal models is critically important and represents the right long-term direction. As noted by Liang et al. (2023), different modalities can exhibit synergy, effectively supporting one another. Madaan et al. (2024) show that multimodal learners often outperform their unimodal counterparts. Similarly, Wang et al. (2024a) argue that modalities benefit each other by helping the model triangulate a shared underlying reality.

Regarding whether performance improves or degrades on benchmarks after our omni-modal training, we observe substantial gains in video understanding (as shown in Table 2, VideoMME w/o subtitles 61.67->63.76, +2.09) and comparable performance on most image (Table 6) and audio tasks (Table 4) relative to single-modality baselines, without increasing model size to accommodate these additional modalities. We expect that scaling the model to three times its current size would yield further performance improvements on various benchmarks of different modalities.

Furthermore, we evaluate the performance of the model checkpoint following modality-specific training and compare it with OmniVinci after omni-modal joint-tuning. As shown in Table 21, omni-modality joint tuning enhances performance across most image and audio tasks.

Table 21: Comparison of model performance between Modality-Specific Training and OmniVinci (After Omni-Modality Tuning) across various benchmarks.

| Benchmark | VQAv2 val | RWQA | SEED Image | MMMU val | ChartQA | AI2D | MathVista | DocVQA val | VideoMME w/o Sub | Omni bench | Daily omni | World sense | LS clean | LS other |
|---|---|---|---|---|---|---|---|---|---|---|---|---|---|---|
| Modality-Specific Training | 79.19 | 64.71 | 73.97 | 45.22 | 77.92 | 87.89 | 52.6 | 88.86 | 59.52 | 31.32 | 56.64 | 44.20 | 1.7 | 4.0 |
| After Omni-Modality Tuning | 83.9 | 67.5 | 77.1 | 49.67 | 84.6 | 91.5 | 63.5 | 92.9 | 68.15 | 46.47 | 66.50 | 48.23 | 1.7 | 3.7 |

## E.6 EVALUATION ON MORE VIDEO BENCHMARKS

Table 22: TOMATO Benchmark Results (Shangguan et al., 2024)

| Model | Category Scores | | | | | | All |
|---|---|---|---|---|---|---|---|
| | Model Rotation | Direction | Velocity & Frequency | Shape & Trend | Visual Cues | Count | |
| VideoLLaMA 2 7B (Cheng et al., 2024b) | 10.10 | 22.80 | 15.70 | 18.80 | 31.40 | 19.50 | 18.50 |
| Phi 3.5 Vision (Abdin et al., 2024) | 20.30 | 16.60 | 14.30 | 23.30 | 40.00 | 24.70 | 20.70 |
| InternVL 2 8B (Chen et al., 2024b) | 17.10 | 25.10 | 09.00 | 28.70 | 31.40 | 22.90 | 21.70 |
| LLaVA-NeXT-Video-32B (Zhang et al., 2024a) | 20.60 | 26.30 | 12.40 | 24.20 | 30.00 | 24.30 | 22.70 |
| InternVL 2 26B (Chen et al., 2024b) | 18.50 | 29.30 | 10.50 | 31.40 | 11.40 | 25.70 | 23.30 |
| VideoLLaMA 2 72B (Cheng et al., 2024b) | 14.30 | 24.60 | 22.40 | 26.50 | 27.10 | 28.80 | 23.50 |
| Video LLaVA 7B (Lin et al., 2023) | 29.40 | 17.90 | 27.10 | 23.30 | 34.30 | 20.90 | 23.60 |
| LLaVA-Video-7B-Video-Only (Zhang et al., 2024d) | 15.40 | 24.10 | 19.50 | 31.40 | 38.60 | 25.70 | 23.90 |
| OmniVinci (ours) | 21.30 | 20.80 | 09.50 | 21.10 | 42.90 | 40.80 | **24.30** |

We further evaluate OmniVinci on two additional benchmarks: TOMATO (Shangguan et al., 2024), which emphasizes video temporal reasoning, and OmniVideoBench (Li et al., 2025a), which focuses on visual–audio synergistic understanding, including long-term temporal reasoning. The quantitative results for both benchmarks are presented in Tables 22 and 23. On TOMATO, OmniVinci achieves competitive performance across diverse temporal reasoning categories, notably excelling in *Visual Cues* and *Count*. On OmniVideoBench, OmniVinci also establishes a new **state-of-the-art overall score among 7B-scale models**, demonstrating robust capability across different video duration ranges. These results collectively highlight OmniVinci's strong generalization ability in complex temporal and multimodal video understanding tasks.

Table 23: OmniVideoBench Benchmark Results (Li et al., 2025a)

| Model | LLM Params | (0,1] min | (1,5] min | (5,10] min | (10,30] min | Overall |
|---|---|---|---|---|---|---|
| VideoLLaMA2 (Cheng et al., 2024b) | 7B | 32.0 | 28.23 | 29.6 | 28.29 | 29.2 |
| Qwen2.5-Omni (7B) (Xu et al., 2025) | 7B | 41.57 | 27.41 | 25.33 | 26.72 | 29.3 |
| Qwen2.5-VL (72B) (Bai et al., 2025) | 72B | 33.13 | 30.03 | 31.88 | 24.43 | 29.5 |
| MiniCPM-o (Yao et al., 2024) | 7B | 31.43 | 28.49 | 34.53 | 26.15 | 29.7 |
| Qwen2.5-VL (7B) (Bai et al., 2025) | 7B | 25.93 | 30.03 | 31.88 | 30.15 | 29.8 |
| HumanOmniV2 (Team, 2025) | 7B | 36.57 | 29.36 | 29.62 | 29.25 | 30.5 |
| Gemini-2.0-Flash (DeepMind, 2025) | – | 33.73 | 35.86 | 32.75 | 22.48 | 31.3 |
| Qwen2.5-VL (32B) (Bai et al., 2025) | 32B | 38.55 | 31.22 | 29.26 | 30.53 | 31.8 |
| OmniVinci (ours) | 7B | 38.55 | 34.11 | 30.13 | 27.16 | **32.1** |

### E.7 TEXT UNDERSTANDING EVALUATION

The primary goal of this work is omni-modal understanding across video, audio, and images, with text-only tasks falling outside its scope. For reference, we evaluate our model on two widely adopted text understanding benchmarks: **MMLU** (Hendrycks et al., 2021) and **MMLU-Pro** (Wang et al., 2024d), in Table 24. These tasks assess a model's ability to reason over a broad range of academic and professional subjects. We compare our system against the closely related Qwen2.5-Omni (Xu et al., 2025). Our model outperforms Qwen2.5-Omni on both benchmarks, with a particularly notable improvement on the more challenging MMLU-Pro dataset, highlighting its enhanced text understanding capabilities.

Table 24: Comparison of model performance on MMLU and MMLU-Pro benchmarks.

| Model | MMLU | MMLU-Pro |
|---|---|---|
| Qwen2.5-Omni | 63.7 | 23.3 |
| OmniVinci | **64.4** | **32.8** |

### E.8 LOSS SUMMARIZATION

We use a weighted sum of the OmniAlignNet loss and cross-entropy loss, defined by the equation:

$$L_{final} = L_{ce} + \alpha L_{o-align}.$$

We determined the optimal value of $\alpha = 0.01$ via an ablation study, the results of which are presented in Table 25.

Table 25: Ablation study results for different values of $\alpha$.

| $\alpha$ | Omnibench | Dailyomni | Worldsense |
|---|---|---|---|
| 0.1 | 44.26 | 65.75 | 45.48 |
| 0.01 | **45.74** | **65.83** | **46.21** |
| 0.001 | 40.12 | 65.67 | 45.90 |

### E.9 ABLATION STUDY OF TEG GROUP DURATION $T_G$

The group duration $T_G$ of Temporal Embeding Grouping (TEG) controls the granularity of the grouping. To examine its impact on model performance, we conduct a series of experiments with varying group durations, as shown in Figure 17. Empirically, we find that a duration of 30 seconds yields the best overall performance. We hypothesize that small $T_G$ may break the semantic coherence between tokens.

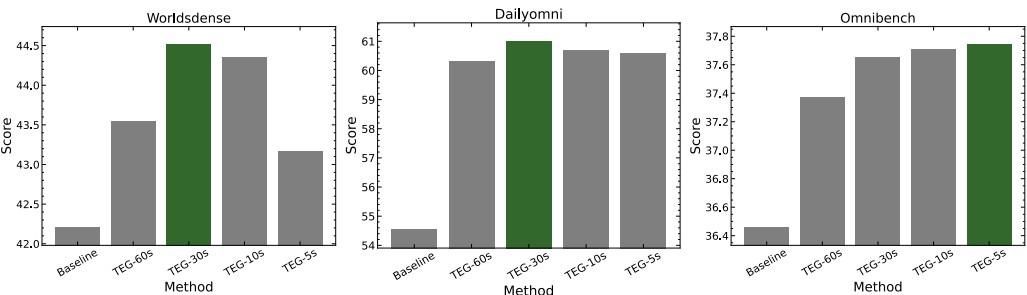

Figure 17: Performance comparison using different TEG group duration $T_G$. We find that a duration of 30 seconds yields the best overall performance.

