# OpenReview forum: "OmniVinci: Enhancing Architecture and Data for Omni-Modal Understanding LLM"
_ICLR.cc/2026/Conference — ICLR 2026 Poster_

### Official Review · Reviewer_uWWz · 2025-11-01

**Soundness:** 3
**Presentation:** 4
**Contribution:** 3
**Rating:** 8
**Confidence:** 4

**Summary:**

This paper introduces OmniVinci, an open-source omni-modal LLM. The core contributions are twofold: (1) A new model architecture featuring three specific mechanisms (OmniAlignNet, TEG, and CRTE) to improve semantic and temporal alignment between vision and audio. (2) A novel data curation pipeline, the "Omni-Modal Data Engine," designed to mitigate "modality-specific hallucination" by using an LLM to correct and fuse single-modality captions. The authors demonstrate that OmniVinci achieves SOTA performance, notably outperforming Qwen2.5-Omni with 6x fewer training tokens.

**Strengths:**

- Novel Alignment System: The paper introduces a novel system for aligning information across text, audio, and video. The proposed architectural modules (OmniAlignNet, TEG, CRTE) are well-motivated and supported by extensive ablation studies (e.g., Table 1).
- Innovative Data Curation: The authors build an "Omni-Modal Data Engine"  to address the common and difficult problem of "modality-specific hallucination." They provide valuable insights (in Figure 4) into the limitations of single-modality captioning .
- State-of-the-Art Performance: The model delivers SOTA results on numerous industry benchmarks for audio, video, and omni-modal tasks among models of similar scale (including Qwen2.5-Omni). Significant improvements are shown on key tests like DailyOmni (+19.05) and Video-MME (+3.9).
- High Training Efficiency: The model achieves this strong performance while being highly efficient, using only 0.2T training tokens compared to Qwen2.5-Omni's 1.2T

**Weaknesses:**

- Limited Gains from RL Post-Training: The performance improvement from the GRPO reinforcement learning (RL) post-training appears relatively modest. As shown in Table 8, the score improvements on Worldsense, Dailyomni, and Omnibench are all less than 1 percentage point. Given the complexity and cost of RL training, does this minor gain suggest a bottleneck in the current method or data?

**Questions:**

- On the Combination of Loss Functions: The OmniAlignNet module introduces a contrastive loss, $L_{o-align}$ (Eq 1)20, while the LLM backbone uses a standard generative cross-entropy loss (implied in Figure 2). How are these (and potentially other) losses combined during the omni-modal joint training phase? Are they simply summed, or are they weighted (e.g., $L_{total} = \alpha L_{LLM} + \beta L_{o-align}$)? If weighted, how were these hyperparameters determined?
- Regarding Modality Conflict at Inference: The data engine is designed to resolve modality-specific hallucination during training . How does the final OmniVinci model handle new, explicit modality conflicts at inference time? For example, if the model is fed a video showing a dog but the audio narration says, "this is a cat," how does the model prioritize or fuse these contradictory signals?
- Clarification on 6x Training Efficiency: The 6x reduction in training tokens (0.2T vs. 1.2T)  is a very impressive efficiency claim. To fully contextualize this: (a) Does the 0.2T token count include the computational cost of running the "Omni-Modal Data Engine" to generate the 24M samples? (b) What was the total computational cost (e.g., total GPU-hours) and wall-clock time required for the omni-modal joint training phase, compared to the baseline?

---

> ### Author Response · Authors · 2025-11-24
> **First Response**
>
> Thank you for your feedback. We will address all of your questions.
>
> > "Limited Gains from RL Post-Training: The performance improvement from the GRPO reinforcement learning (RL) post-training appears relatively modest."
>
> As shown in Table 8, omni-modal RL post-training consistently improves performance across all omni-modal benchmarks. On OmniBench specifically, we observe a gain of +1.32. Considering the RL dataset contains only 18k samples, the training process is quite efficient (~160 GPU hours). We plan to expand this dataset to further enhance model performance in the future.
>
> > "On the Combination of Loss Functions: The OmniAlignNet module introduces a contrastive loss,  (Eq 1)20, while the LLM backbone uses a standard generative cross-entropy loss (implied in Figure 2). How are these (and potentially other) losses combined during the omni-modal joint training phase? Are they simply summed, or are they weighted (e.g., )? If weighted, how were these hyperparameters determined?"
>
> We use a weighted sum of the OmniAlignNet loss and cross-entropy loss, defined by the equation: $L_{final} = L_{ce} + \alpha L_{o-align}$. We decided the value of $\alpha=0.01$ via ablation study:
>
> | $\alpha$      | Omnibench | Dailyomni | Worldsense |
> |---------|-----------|-----------|------------|
> | 0.1     | 44.26     | 65.75     | 45.48      |
> | 0.01    | **45.74** | **65.83** | **46.21**  |
> | 0.001   | 40.12     | 65.67     | 45.90      |
>
>
> We have added this discussion to Section E.8 in Appendix.
>
>
> > "Regarding Modality Conflict at Inference: The data engine is designed to resolve modality-specific hallucination during training . How does the final OmniVinci model handle new, explicit modality conflicts at inference time? For example, if the model is fed a video showing a dog but the audio narration says, "this is a cat," how does the model prioritize or fuse these contradictory signals?"
>
>
> This is an interesting question related to potential data poisoning. As suggested by the reviewer, we present OmniVinci with a video of a dog paired with audio stating, 'this is a cat'. The model consistently respond 'It's a dog'. This suggests that the model relies primarily on visual cues when facing contradictory audio-visual inputs. We intend to explore this phenomenon more deeply in future studies.
>
>
> > "Clarification on 6x Training Efficiency: The 6x reduction in training tokens (0.2T vs. 1.2T) is a very impressive efficiency claim. To fully contextualize this: (a) Does the 0.2T token count include the computational cost of running the "Omni-Modal Data Engine" to generate the 24M samples? (b) What was the total computational cost (e.g., total GPU-hours) and wall-clock time required for the omni-modal joint training phase, compared to the baseline?"
>
> The training token counts indicate the amount of data used for training. The estimated computational cost for generating omni-modal data is roughly 2,000 GPU hours, which is relatively small compared to the overall model training, and the data only needs to be generated once. Since Qwen2.5-Omni did not disclose the compute cost for their data generation, a direct comparison is not possible. However, given that they use six times more data than we do, their costs for data generation and curation are likely significantly higher than us.
> Qwen2.5-Omni did not disclose the compute used for their training. They use six times more training tokens, and their model is larger than OmniVinci, so their training costs are likely much higher than ours. The estimated training cost for our model is approximately 40,000 GPU hours on NVIDIA A100 GPUs.
>
> We hope the new experimental results and analysis help clarify and strengthen the understanding of our work. Please feel free to let us know if you have any remaining concerns that might prevent you from increasing the paper’s rating.

---

> > ### Comment · Reviewer_uWWz · 2025-11-27
> > **Response to author**
> >
> > Thank you for your valuable response. I will maintain my original score.

---

### Official Review · Reviewer_xmSv · 2025-11-01

**Soundness:** 3
**Presentation:** 3
**Contribution:** 3
**Rating:** 6
**Confidence:** 4

**Summary:**

This paper introduces OmniVinci, an omni-modal Large Language Model (LLM) designed for comprehensive cross-modal understanding by jointly processing vision, audio, and language. The authors present notable architectural contributions, including OmniAlignNet for vision-audio alignment, Temporal Embedding Grouping (TEG) for structured temporal representation of modality tokens, and Constrained Rotary Time Embedding (CRTE) for encoding absolute temporal information. They also curated and synthesized a diverse, large-scale dataset comprising 24 million conversations spanning both single- and multi-modal scenarios. Commendably, the work includes demonstrations of practical applications in real-world settings.

**Strengths:**

- The work proposes the innovative multi-modal integration architecture, OmniAlignNet, which aligns image and video dimensions within a unified feature space. The introduction of TEG and CRTE further enhances modality feature learning, substantially boosting the model's overall omni-modal understanding performance.

- The work construct a substantial dataset of 24 million samples and implemented multi-modal reasoning augmentation. The workflow illustrat in Figure 4 provides a clear mechanism for handling modality-specific hallucinations and generating high-quality cross-modal supervision.

- The paper includes a relatively comprehensive set of evaluations and ablation studies, effectively demonstrating the model's capabilities. Furthermore, the work showcases initial deployment and application potential in real-world scenarios.

**Weaknesses:**

- The study employs a paradigm where single modalities (image, audio) are trained separately before a unified modal alignment is performed. The paper lacks a performance comparison of modality-specific tasks before and after the cross-modal unified alignment. Reporting the respective performances on pure Image and Audio tasks before and after this alignment stage would significantly help validate the effectiveness of the proposed method.

- The image data significantly outweighs the audio data during pre-training. I wonder what the ratio of image-to-audio modality tokens is during the cross-modal alignment phase. Moreover, if the compression ratio for both the vision and audio encoders is identical during alignment,  how to eliminate potential bias resulting from the inherent token quantity imbalance between these modalities.

- Some reported model results are not reflective of the current state-of-the-art. It would be beneficial to use more recent, cutting-edge model results for comparison (e.g., updating InternVL-2 to InternVL-3 and Qwen2-vl to Qwen3-vl) to ensure the novelty claims are appropriately contextualized against the strongest contemporaries.

**Questions:**

same as weakness

---

> ### Author Response · Authors · 2025-11-24
> **First Response**
>
> Thank you for your feedback. We will respond to each question individually.
>
> > "The study employs a paradigm where single modalities (image, audio) are trained separately before a unified modal alignment is performed. The paper lacks a performance comparison of modality-specific tasks before and after the cross-modal unified alignment. Reporting the respective performances on pure Image and Audio tasks before and after this alignment stage would significantly help validate the effectiveness of the proposed method."
>
> As suggested by the reviewer, we evaluate the checkpoint after modality-specific training, and compare it with OmniVinci (after omni-modal joint-turning), and show the results in the table below.
>
> | Benchmark                  | VQAv2-val | RealWorldQA | SEED-Image | MMMU-val | ChartQA | AI2D  | MathVista | DocVQA-val | VideoMME w/o Sub | Omnibench | Dailyomni | Worldsense | LS-clean | LS-other |
> |----------------------------|-----------|-------------|------------|----------|---------|-------|-----------|------------|-----------------|-----------|-----------|------------|----------|----------|
> | Modality-Specific Training | 79.19     | 64.71       | 73.97      | 45.22    | 77.92   | 87.89 | 52.6      | 88.86      | 59.52           | 31.32     | 56.64     | 44.20      | **1.7** | 4.0     |
> | After Omni-Modality Tuning                  | **83.9**  | **67.5**    | **77.1**   | **49.667** | **84.6** | **91.5** | **63.5**  | **92.9**   | **68.15**       | **46.47** | **66.50** | **48.23** | **1.7**     | **3.7** |
>
>
> According to the results, omni-modality joint tuning enhances performance across most image and audio tasks. This improvement may be attributed to omni-modality training further strengthening the model's capability to understand and reason over multimodal signals. We have added this discussion to Section E.5 of the paper.
>
>
>
> > "The image data significantly outweighs the audio data during pre-training. I wonder what the ratio of image-to-audio modality tokens is during the cross-modal alignment phase. Moreover, if the compression ratio for both the vision and audio encoders is identical during alignment, how to eliminate potential bias resulting from the inherent token quantity imbalance between these modalities."
>
> As described in Section 3.2, image-text data accounts for approximately 36% of the overall omni-modal training mixture, while audio-text data represents around 38%. Consequently, the resulting training tokens for these two modalities are similar in volume.
>
>
> > "Some reported model results are not reflective of the current state-of-the-art. It would be beneficial to use more recent, cutting-edge model results for comparison (e.g., updating InternVL-2 to InternVL-3 and Qwen2-vl to Qwen3-vl) to ensure the novelty claims are appropriately contextualized against the strongest contemporaries."
>
> Thank you for your suggestions. We have updated Table 5 and 6 of the paper with InternVL-3 and Qwen3-VL results on different visual benchmarks.
>
> We believe the updated experiments and analysis offer clearer insight into our contribution. The discussion has been incorporated into the submission. Please let us know if anything else remains unclear or could stand in the way of a higher rating.

---

### Official Review · Reviewer_Tky5 · 2025-11-01

**Soundness:** 3
**Presentation:** 2
**Contribution:** 2
**Rating:** 4
**Confidence:** 2

**Summary:**

This paper introduces a suite of modules to enhance video–audio multimodal understanding, including OmniAlignNet, Temporal Embedding Grouping (TEG), and Constrained Rotary Time Embedding (CRTE).
OmniAlignNet is proposed to align video and audio latent representations. TEG and CRTE are designed to improve the temporal alignment of video and audio features, thereby facilitating model learning.
Ablation studies validate the effectiveness of each component. The proposed method outperforms Qwen2.5-Omni on several video and audio understanding benchmarks.

**Strengths:**

- Experiments on dataset engines highlight the importance of fully exploiting cross-modal information in sounding videos, which benefits both clean data construction and model performance. These findings offer meaningful insights for future research.
- The proposed modules improve model performance from two key perspectives—video–audio semantic alignment and temporal alignment—both of which are well-motivated and demonstrated to be effective for video and audio understanding tasks.

**Weaknesses:**

- Although the authors claim the model is omni-modal, the work primarily focuses on video and audio, leading to degraded performance on the image modality. Furthermore, results for text-to-text tasks are not reported. Combined with the claim that the model uses far more tokens than Qwen2.5-Omni, it raises concerns that the proposed approach may neglect text and image modalities.
- TEG involves interleaving video and audio tokens when feeding the LLM, which is similar to the approach used in Qwen2.5-Omni. Likewise, contrastive learning is a common practice for aligning video and audio semantic features.
- Comparisons with Qwen2.5-Omni in terms of audio generation latency, training cost, and total parameter count are not reported.

**Questions:**

Since the 24M multimodal model is part of the contributions, is there a plan to open-source it?

---

> ### Author Response · Authors · 2025-11-24
> **First Response (1/n)**
>
> Thank you for your feedback. We will respond to all your questions.
>
> > "The work primarily focuses on video and audio, leading to degraded performance on the image modality."
>
> For image understanding, our model uses a training data mixture similar to NVILA [1], making it the most comparable baseline. As shown in Table 6, OmniVinci delivers performance on par with NVILA for image tasks and even surpasses it in some cases (e.g., on SEED Image and VQAv2). More importantly, the new OmniVinci model processes video, audio, and images using a single shared parameter set. We do not increase the parameter count to support multiple modalities, allowing for a fairer comparison with state-of-the-art single-task models. Under this constraint, OmniVinci’s superior performance in omni-modal and video understanding—while still maintaining competitive results on image- and audio-specific tasks—is particularly notable.
>
> > "Results for text-to-text tasks are not reported. Combined with the claim that the model uses far more tokens than Qwen2.5-Omni, it raises concerns that the proposed approach may neglect text and image modalities."
>
> To clarify, our method in fact uses far **fewer** training tokens (1/6) than Qwen2.5-Omni while delivering stronger performance on many tasks, demonstrating its training efficiency. The primary goal of this work is multimodal understanding across video, audio, and images, with text-only tasks falling outside its scope.
>
> We next report **MMLU** and **MMLU-Pro** results and compare it against Qwen2.5-Omni in the table below. Our model outperforms Qwen2.5-Omni on both benchmarks, with a particularly notable improvement on the more challenging MMLU-Pro dataset, highlighting its enhanced text understanding capabilities.
>
>
> | Model          | MMLU | MMLU-Pro |
> |----------------|------|----------|
> | Qwen2.5-Omni   | 63.7 | 23.3     |
> | **OmniVinci**      | **64.4** | **32.8**     |
>
> We have added these results to Section E.7 of the paper.
>
>
> > "TEG involves interleaving video and audio tokens when feeding the LLM, which is similar to the approach used in Qwen2.5-Omni. Likewise, contrastive learning is a common practice for aligning video and audio semantic features."
>
> There is an important technical difference: Qwen2.5-Omni strictly interleaves multimodal tokens according to their exact timestamps. This is a special case of our Temporal Embedding Grouping where the group duration $T_G$ is small. However, as illustrated in Section E.9 (Figure 17), a small $T_G$ leads to worse model performance because it may break the semantic coherence between tokens and thus be suboptimal. By controlling the duration of each temporal group, we adjust how vision and audio tokens are interleaved in groups that are more favored by MLLM joint training. This enables us to improve semantic coherence between tokens within each modality.
>
> Contrastive learning—such as in CLIP [4]—is adopted for image-text pretraining, but prior work has not explored its use in **post-training** omni-modal LLMs. In this paper, we introduce OmniAlignNet,  the first module for the overall MLLM training that maps video and audio tokens into a unified latent space using cross-attention and enforces alignment between visual and auditory signals. This contrastive objective is jointly optimized with a cross-entropy loss to guide the training of the omni-modal LLM. The improvement is significant, as shown in Table 1.
>
> > "Comparisons with Qwen2.5-Omni in terms of audio generation latency, training cost, and total parameter count"
>
> We appreciate the feedback. Here, we further report an enhanced analysis of consumed training tokens, model parameter sizes, and inference latency across various GPU configurations.
>
> | Model            | Training Tokens | Total Parameters                   | Architecture                     |  Latency (NVIDIA A100 80GB) | Latency (NVIDIA L40S 48GB) | Latency (RTX 4090 24GB) |
> |-----------------|----------------|-----------------------------------|---------------------------------|------------------|------------------|----------------|
> | Qwen2.5-Omni     | 1.2T           | 10.7B (8.9B + 1.8B Talker) | End-to-End Streaming Audio Tokens | ~400 ms          | ~450 ms          | ~500 ms        |
> | OmniVinci (Ours) |  **0.2T**           | **9.15B (8.7B + 0.45B TTS)**    | Cascaded (Agentic) Decoupled Synthesis | **~350 ms**          | **~380 ms**          | **~420 ms**        |
>
> Our OmniVinci demonstrates clearly superior efficiency.
>
> > "Since the 24M multimodal model is part of the contributions, is there a plan to open-source it?"
>
> Yes — we will open-source the data. Our code and model have already received over **15,000+** downloads to date. The core goal of this work is to help accelerate open-source advancements in omni-modal LLM development.
>
> The newly added experiments and analysis should help clarify our approach and strengthen its presentation. We would be glad to address any other issues that might affect your evaluation.

---

> ### Author Response · Authors · 2025-11-24
> **First Response (2/n)**
>
> References:
>
> [1] Liu et al., NVILA: Efficient frontier visual language models. CVPR 2025
>
> [2] Hendrycks et al., Measuring Massive Multitask Language Understanding. ICLR 2021
>
> [3] Wang et al., MMLU-Pro: A More Robust and Challenging Multi-Task Language Understanding Benchmark. NeurIPS 2024
>
> [4] Radford et al., Learning transferable visual models from natural language supervision, ICML 2021

---

### Official Review · Reviewer_hewe · 2025-11-02

**Soundness:** 2
**Presentation:** 2
**Contribution:** 2
**Rating:** 2
**Confidence:** 4

**Summary:**

The paper introduces OmniVinci with three techniques including i) OmniAlignNet to align vision and audio in a video ii) Temporal embedding grouping for capturing the temporal alignment between vision and audio and iii) Constrained rotary time embedding for adding temporal information into vision-audio embeddings. The experiments show improvement on various multimodal understanding, audio understanding and vision understanding tasks over prior work.

**Strengths:**

The paper tackles an important goal of building open-source, omni-modal LLM by incorporating OmniAlignNet, temporal embedding grouping and constrained rotary time embedding. The paper curated a large-scale dataset spanning audio, video and image domains and shows improvements to multimodal understanding, audio understanding and vision understanding tasks over prior work, which would be of interest to the community. Overall, the proposed method is simple and paper is easy to read and well-written.

**Weaknesses:**

* The distinction of OmniAlignNet module, use of position encoding from the current video-audio alignment common in existing work [1,2,3] is unclear. The paper lacks a discussion with these works making its positioning among them unclear.
* Similarly there exists many studies [4,5] that have incorporated the temporal sequence in multiple ways, which the paper lacks a comparison or distinction with.
* The paper argues the need of Omnimodal data engine and gives an example of where both audio and video are required. But as shown in many prior multimodal studies [6,7,8], there exists many datasets where one modality suffices and thus explicitly enforcing interactions is suboptimal and often leads to unnecessary correlations. The paper lacks any discussion in this aspect as well.
* The paper highlights modality-specific training in section 3.1 by using data for each modality but it is unclear how this is incorporated in the omni-modal joint training and more details need to be provided on the separation of the modality-specific and omni-modal training.
* The claims of the paper are not well supported by the empirical results. For example, i) While OmniVinci only improves the performance on Dailyomni in Table 3, its worse than almost all models on Omnibench with upto 10% worse than Qwen. The performance on Worldsense is also not convincing without confidence intervals. Similar conclusions hold for Image benchmarks in Table 7 and speech recognition benchmarks in Table 5, where OmniVinci obtains worse performance across baselines.

Overall, in the current state I recommend recommend rejection due to the lack of discussion with prior work in multiple aspects and claims not being supported by empirical results.

The following can improve the paper further:
* A common trend for multimodal models is the lack of temporal reasoning. It would be useful to see the performance of the proposed method on cases [9,10] which are explicitly designed to evaluate the same.
* The font size and presentation for most results is extremely small, which makes it challenging to interpret the results meaningfully.
* The position of Table 5 and Table 6 can be switched.

References:
[1] Cheng et al. MMAudio: Taming Multimodal Joint Training for High-Quality Video-to-Audio Synthesis.
[2] Kim et al. Deep Visual Forced Alignment: Learning to Align Transcription with Talking Face Video.
[3] Guo et al. Aligned Better, Listen Better for Audio-Visual Large Language Models.
[4] Zerveas et al.  A transformer-based framework for multivariate time series representation learning.
[5] Eldele et al. TSLANet: Rethinking Transformers for Time Series Representation Learning.
[6] Liang et al. Quantifying & Modeling Multimodal Interactions: An Information Decomposition Framework.
[7] Madaan et al. Jointly Modeling Inter- & Intra-Modality Dependencies for Multi-modal Learning.
[8] Wang et al. An Information Criterion for Controlled Disentanglement of Multimodal Data.
[9] Shangguan et al. TOMATO: Assessing Visual Temporal Reasoning Capabilities in Multimodal Foundation Models.
[10] Video SimpleQA: Towards Factuality Evaluation in Large Video Language Models.

**Questions:**

Please refer to my comments above.

---

> ### Author Response · Authors · 2025-11-23
> **First Response (1/n)**
>
> Thanks for the feedback. We next address concerns in a point-by-point manner.
>
> > "The distinction of OmniAlignNet module, use of position encoding from the current video-audio alignment common in existing work [1,2,3] is unclear. Similarly there exists many studies [4,5] that have incorporated the temporal sequence in multiple ways."
>
> Our proposals are very different from the works suggested amid differences in objectives and difficulty levels for multimodal perception to perceive all modalities all at once. We have added the discussion about these works to the Related Work part (Section A) in our paper, and provide a detailed analysis next:
>
>
> - Reference [1] concentrates exclusively on **audio generation**, whereas our work targets multimodal understanding. As a result, the underlying motivation and technical design differ substantially.
>
>
>
> - Reference [2] **does not handle audio at all**. It focuses solely on vision–text alignment and does not address vision–audio alignment. It trains a neural network to predict the text transcription for each frame, which is quite different from the scope of our paper.
>
>
> - Reference [3] combines visual and audio tokens of the same timestamp through cross attention. It doesn’t insert the absolute temporal information into the tokens. Differently, our method inserts **absolute temporal information** through embedding rotation in CRTE.
>
> - References [4] and [5] focus on the task of **time series analysis**. No vision or audio modalities are involved.
>
> We have added the discussion about all of these works to the Related Work part (Section A) in our paper.
>
>
> > "The paper argues the need of Omnimodal data engine and gives an example of where both audio and video are required. But as shown in many prior multimodal studies [6,7,8], there exists many datasets where one modality suffices and thus explicitly enforcing interactions is suboptimal and often leads to unnecessary correlations."
>
>
> We respectfully disagree that learning harmony across harmony modalities is always suboptimal - as for humans, we leverage multimodal inputs for a joint informative decision. In fact, as noted in [6], different modalities can exhibit synergy, effectively supporting one another. [7] shows that multimodal learners often outperform their unimodal counterparts. Similarly, [8] argues that modalities benefit each other by helping the model triangulate a shared underlying reality.
>
> Meanwhile, we observed that some academic benchmarks may rely on a single modality and exhibit single-dataset bias. This is also the reason why we report **30+** benchmark results across diverse domains to more robustly validate the model and support our ablations.
>
>
> Regarding whether performance is improved on single-modality benchmarks after our omni-modal training, we evaluate our checkpoint after modality-specific training, and compare it with OmniVinci (after omni-modal joint-turning), and show the results in the table below.
>
> | Benchmark                  | VQAv2-val | RealWorldQA | SEED-Image | MMMU-val | ChartQA | AI2D  | MathVista | DocVQA-val | VideoMME w/o Sub | Omnibench | Dailyomni | Worldsense | LS-clean | LS-other |
> |----------------------------|-----------|-------------|------------|----------|---------|-------|-----------|------------|-----------------|-----------|-----------|------------|----------|----------|
> | Modality-Specific Training | 79.19     | 64.71       | 73.97      | 45.22    | 77.92   | 87.89 | 52.6      | 88.86      | 59.52           | 31.32     | 56.64     | 44.20      | **1.70** | 4.04     |
> | After Omni-Modality Tuning                  | **83.9**  | **67.5**    | **77.1**   | **49.667** | **84.6** | **91.5** | **63.5**  | **92.9**   | **68.15**       | **46.47** | **66.50** | **48.23** | **1.70**     | **3.70** |
>
> According to the results, omni-modality joint tuning enhances performance across most image and audio tasks. This improvement may be attributed to the fact that omni-modality training further strengthens the model's capability to understand and reason over multimodal signals. We expect that tripling the model’s size, and thus enhancing its capacity for all three modalities, will result in further improvements across diverse multimodal benchmark tasks.
>
> What’s more important, humans perceive the world through vision and audio simultaneously and rely on the correlation of different modalities to perform many tasks. Developing omni-modal models is important and represents the right long-term direction, as also acknowledged in the Strengths section by the same reviewer.
>
>
> We have added this discussion to Section E.5 of the paper.

---

> ### Author Response · Authors · 2025-11-23
> **First Response (2/n)**
>
> > "The paper highlights modality-specific training in section 3.1 by using data for each modality but it is unclear how this is incorporated in the omni-modal joint training and more details need to be provided on the separation of the modality-specific and omni-modal training."
>
> We will open-source all training code, setup, recipes, and checkpoints of OmniVinci for full reproducibility. Note that these are training steps that yield one single final foundation OmniVinci model that perceives all modalities. To further elaborate on the carefully designed recipes as described in the first paragraph of Section 3 (starting from Line 210), we use a two-stage approach for training: we first conduct modality-specific training to develop individual capabilities for each modality, followed by omni-modal joint training to integrate these capabilities. For modality-specific training (Section 3.1), we provide more details in **Appendix D.3** because of the page limit. For omni-modal joint training, we detail the training data in Section 3.2 and **Appendix D.5**, and introduce training hyperparameters in **Appendix D.4**.
>
>
> > "i) While OmniVinci improves the performance on Dailyomni in Table 3, its worse than almost all models on Omnibench with up to 10% worse than Qwen. The performance on Worldsense is also not convincing without confidence intervals."
>
>
> We unbiasedly profile across popular omni benchmarks and report all our results. Among the omni-modal benchmarks we examined, Dailyomni and Worldsense focus on video-audio understanding, whereas Omnibench evaluates the less usual image-audio setting that our current recipe contains zero such modality-pair training instances. Notice that OmniVinci already delivers strong performance across 30+ benchmarks across single and omni domains with one model, with more potential on selective datasets such as OmniBench to future work. Notably, our model surpasses Qwen2.5-Omni by **+19.05%** and Worldsense by **+2.83%** in video-audio omni-modal understanding, which are very significant gains. For Worldsense, to verify the confidence interval, we repeated the evaluation three times using a generation sampling temperature of 0.7 and observed a deviation of only 0.2548, confirming that the **+2.83% improvement is statistically significant**.
>
> We further benchmark our model on another omni-modality benchmark, OmniVideoBench, and present those results in the answer to the next question. Notably, OmniVinci sets a new **state-of-the-art** overall score among models smaller than 10B. Notably, the OmniVideoBench team conducted this assessment using our open-source model weights, further confirming the model’s comprehensive omni-modal understanding capabilities.

---

> ### Author Response · Authors · 2025-11-23
> **First Response (3/n)**
>
> > "A common trend for multimodal models is the lack of temporal reasoning. It would be useful to see the performance of the proposed method on cases which are explicitly designed to evaluate the same."
>
> As suggested we evaluate **OmniVinci** on two additional benchmarks: **TOMATO** [9] and **OmniVideoBench** [10]. The results are shown in the tables below.
>
> ### **TOMATO**
>
> | Model           | ModelRotation (286) | Direction (403) | Velocity & Frequency (210) | Shape & Trend (223) | Visual Cues (70) | Count (292) | All (1,484) |
> |-------------------------------|------------------|----------------|---------------------------|-------------------|-----------------|-------------|--------------|
> | VideoLLaMA 2 7B             | 10.1             | 22.8           | 15.7                      | 18.8              | 31.4            | 19.5        | 18.5         |
> | Phi 3.5 Vision                 | 20.3             | 16.6           | 14.3                      | 23.3              | 40.0            | 24.7        | 20.7         |
> | InternVL 2 8B                  | 17.1             | 25.1           | 9.0                       | 28.7              | 31.4            | 22.9        | 21.7         |
> | LLaVA-NeXT-Video-32B           | 20.6             | 26.3           | 12.4                      | 24.2              | 30.0            | 24.3        | 22.7         |
> | InternVL 2 26B                 | 18.5             | 29.3           | 10.5                      | 31.4              | 11.4            | 25.7        | 23.3         |
> | VideoLLaMA 2 72B             | 14.3             | 24.6           | 22.4                      | 26.5              | 27.1            | 28.8        | 23.5         |
> | Video LLaVA 7B               | 29.4             | 17.9           | 27.1                      | 23.3              | 34.3            | 20.9        | 23.6         |
> | LLaVA-Video-7B-Video-Only      | 15.4             | 24.1           | 19.5                      | 31.4              | 38.6            | 25.7        | 23.9         |
> | OmniVinci                      | 21.3             | 20.8           | 9.5                       | 21.1              | 42.9            | 40.8        | **24.3**         |
>
>
> ### **OmniVideoBench**
>
> | Model                    | LLM Params | (0,1] min | (1,5] min | (5,10] min | (10,30] min | Overall |
> | ------------------------ | ------ | ---------- | ---------- | ---------- | ------------ | ------- |
> | VideoLLaMA2              | 7B     | 32.0       | 28.23      | 29.6       | 28.29        | 29.2    |
> | Qwen2.5-Omni (7B)        | 7B     | 41.57      | 27.41      | 25.33      | 26.72        | 29.3    |
> | Qwen2.5-VL (72B)         | 72B    | 33.13      | 30.03      | 31.88      | 24.43        | 29.5    |
> | MiniCPM-o                | 7B     | 31.43      | 28.49      | 34.53      | 26.15        | 29.7    |
> | Qwen2.5-VL (7B)          | 7B     | 25.93      | 30.03      | 31.88      | 30.15        | 29.8    |
> | HumanOmniV2              | 7B     | 36.57      | 29.36      | 29.62      | 29.25        | 30.5    |
> | Gemini-2.0-Flash         | –      | 33.73      | 35.86      | 32.75      | 22.48        | 31.3    |
> | Qwen2.5-VL (32B)         | 32B    | 38.55      | 31.22      | 29.26      | 30.53        | 31.8    |
> | OmniVinci                | 7B     | 38.55      | 34.11      | 30.13      | 27.16        | **32.1**    |
>
>
> On TOMATO, our model achieves significantly better performance than strong video-only LLMs (LLaVA-NeXT-Video-32B, VideoLLaMA 2 72B, InternVL 2 8B, etc.)  across diverse temporal reasoning categories, notably excelling in Visual Cues and Count.
>
> On OmniVideoBench, OmniVinci also establishes a **new state-of-the-art overall score among 7B-scale models**. Especially, this evaluation is conducted by the OmniVideoBench team using our open-source model weights, validating the model’s omni-modal understanding capabilities.
>
> These results collectively highlight our model’s strong generalization ability in complex temporal and omni-modal video understanding tasks.
>
> We have updated these results in Section E.6 of the Appendix.
>
> > "The font size and presentation for most results is extremely small"
>
> We observed this challenge due to the page limit. We will increase the font in the revised manuscript.
>
>
> > "The position of Table 5 and Table 6 can be switched."
>
>
> We have adjusted the location of these two tables.
>
>
> The additional experiments and analysis were added to provide further clarity on our work. If there are still any concerns that could influence the paper’s rating, we are happy to address them.

---

> ### Author Response · Authors · 2025-11-23
> **First Response (4/n)**
>
> We have addressed the questions raised in the review. We are happy to further elaborate if you have any further questions about this submission.
>
> References:
>
> [1] Cheng et al. MMAudio: Taming Multimodal Joint Training for High-Quality Video-to-Audio Synthesis.
>
> [2] Kim et al. Deep Visual Forced Alignment: Learning to Align Transcription with Talking Face Video.
>
> [3] Guo et al. Aligned Better, Listen Better for Audio-Visual Large Language Models.
>
> [4] Zerveas et al. A transformer-based framework for multivariate time series representation learning.
>
> [5] Eldele et al. TSLANet: Rethinking Transformers for Time Series Representation Learning.
>
> [6] Liang et al. Quantifying & Modeling Multimodal Interactions: An Information Decomposition Framework.
>
> [7] Madaan et al. Jointly Modeling Inter- & Intra-Modality Dependencies for Multi-modal Learning.
>
> [8] Wang et al. An Information Criterion for Controlled Disentanglement of Multimodal Data.
>
> [9] Shangguan et al. TOMATO: Assessing Visual Temporal Reasoning Capabilities in Multimodal Foundation Models.
>
> [10] Chaorui et al. OmniVideoBench: Towards Audio-Visual Understanding Evaluation for Omni MLLMs

---

### Meta-Review · Area_Chair_ZvDP · 2026-01-05

**Summary:**

This paper proposes OmniVinci, an open-source omni-modal LLM that jointly processes vision, audio, and text. The key contributions include three architectural components: OmniAlignNet, Temporal Embedding Grouping (TEG), and Constrained Rotary Time Embedding (CRTE), as well as a large-scale omni-modal data curation pipeline producing 24M training samples.

Reviewers generally agree that the paper addresses an important problem, presents technically sound methods, and demonstrates strong empirical performance, particularly in video–audio omni-modal understanding and training efficiency.

The main concerns raised by reviewers centered on (i) lack of discussion to prior work, (ii) details on the training recipe, and (iii) missing baselines. The authors provided a detailed rebuttal, including new experiments, additional benchmarks, ablations, and clarifications . Overall, the rebuttal substantially strengthens the paper and resolves most substantive concerns.

AC's recommendation for this paper is Accept (Poster)

**Reviewer Concerns:**

Most substantive concerns raised during the review were addressed in the rebuttal, particularly for the two reviewers (hewe and Tky5) who initially gave negative scores.

Reviewer hewe

Concern 1: The three main contributions (OmniAlignNet, Temporal Embedding Grouping, and Constrained Rotary Time Embedding) lacked sufficient discussion and fundamental differentiation from prior work.

Status: Partially addressed. The rebuttal provides reasonable clarifications and highlights practical differences from the cited work; however, several responses are application- or practice-driven, whereas the reviewer’s concern was more fundamental.

Concern 2: The paper’s key claims were not sufficiently supported by empirical results.

Status: Addressed. The authors added substantial new experiments and benchmarks, which significantly strengthen empirical support for the main claims.

Reviewer Tky5

Concern: Missing or insufficient experimental evidence on several aspects of the model and comparisons.

Status: Addressed. The authors added the requested experiments and analyses in the rebuttal, resolving the reviewer’s concerns.

**Reviewer Scores:**

I would expect reviewers hewe and Tky5 to increase their scores toward to a borderline/weak-accept range, while reviewers xmSv and uWWz would likely keep their scores unchanged.

---

### Decision · Program_Chairs · 2026-01-26

Accept (Poster)